# Differentiable Cyclic Causal Discovery Under Unmeasured Confounders

**Muralikrishnna G. Sethuraman**
School of Electrical & Computer Engineering
Georgia Institute of Technology
muralikgs@gatech.edu

**Faramarz Fekri**
School of Electrical & Computer Engineering
Georgia Institute of Technology
faramarz.fekri@ece.gatech.edu

## Abstract

Understanding causal relationships between variables is fundamental across scientific disciplines. Most causal discovery algorithms rely on two key assumptions: (i) all variables are observed, and (ii) the underlying causal graph is acyclic. While these assumptions simplify theoretical analysis, they are often violated in real-world systems, such as biological networks. Existing methods that account for confounders either assume linearity or struggle with scalability. To address these limitations, we propose DCCD-CONF, a novel framework for differentiable learning of nonlinear cyclic causal graphs in the presence of unmeasured confounders using interventional data. Our approach alternates between optimizing the graph structure and estimating the confounder distribution by maximizing the log-likelihood of the data. Through experiments on synthetic data and real-world gene perturbation datasets, we show that DCCD-CONF outperforms state-of-the-art methods in both causal graph recovery and confounder identification. Additionally, we provide consistency guarantees for our framework, reinforcing its theoretical soundness.

## 1 Introduction

Modeling cause-effect relationships between variables is a fundamental problem in science [1, 2, 3], as it enables the prediction of a system's behavior under previously unseen perturbations. These relationships are typically represented using *directed graphs* (DGs), where nodes correspond to variables, and directed edges capture causal dependencies. Consequently, causal discovery reduces to learning the structure of these graphs.

Existing causal discovery algorithms can be broadly classified into three categories: (i) constraint-based methods, (ii) score-based methods, and (iii) hybrid methods. Constraint-based methods, such as the PC algorithm [4, 5, 6], search for causal graphs that best satisfy the independence constraints observed in the data. However, since the number of conditional independence tests grows exponentially with the number of nodes, these methods often struggle with scalability. Score-based methods, such as the GES algorithm [7, 8], learn graph structures by maximizing a penalized score function, such as the Bayesian Information Criterion (BIC), over the space of graphs. Given the vast search space, these methods often employ greedy strategies to reduce computational complexity. A significant breakthrough came with Zheng et al. [9], who introduced a continuous constraint formulation to restrict the search space to acyclic graphs, inspiring several extensions [10, 11, 12, 13, 14, 15] that frame causal discovery as a continuous optimization problem under various model assumptions. Hybrid methods [16, 17, 18] integrate aspects of both constraint-based and score-based approaches, leveraging independence constraints while optimizing a score function.

Most causal structure learning methods assume (i) a *directed acyclic graph* (DAG) with no directed cycles and (ii) complete observability, meaning no unmeasured confounders.

While these assumptions simplify the search space, they are often unrealistic, as real-world systems—especially in biology—frequently exhibit feedback loops and hidden confounders [19]. Enforcing these constraints can also increase computational complexity, particularly in ensuring acyclicity, which often requires solving challenging combinatorial or constrained optimization problems. These limitations hinder the practical applicability of existing methods in settings where such violations are unavoidable.

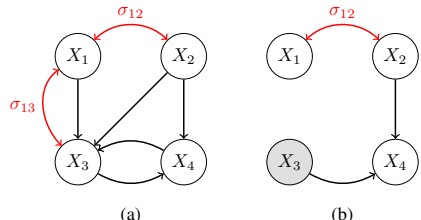

Figure 1: (a) Example of a directed mixed graph $\mathcal{G}$, where the bidirectional edges represent hidden confounders, with $\sigma_{ij}$ indicating their corresponding strengths; (b) Mutilated graph, $\mathrm{do}(I_k)(\mathcal{G})$, resulting from the interventional experiment $I_k = \{X_3\}$, where all incoming edges (including bidirectional edges) to $X_3$ are removed.

Several approaches have been developed to address the challenge of feedback loops within causal graphs. Early work by Richardson [20] extended constraint-based approaches for Directed Acyclic Graphs (DAGs) to accommodate directed cycles. Another key contribution came from Lacerda et al. [21], who generalized Independent Component Analysis (ICA)-based causal discovery to handle linear non-Gaussian cyclic graphs. More recently, a growing body of research has focused on score-based methods for learning cyclic causal graphs [22, 23, 24, 25]. Additionally, some approaches leverage interventional data to improve structure recovery in cyclic systems. For instance, Hyttinen et al. [26] and Huetter and Rigollet [22] introduced frameworks that explicitly incorporate interventions to refine cyclic graph estimation. Sethuraman et al. [27] further advanced this line of research by introducing a differentiable framework for learning nonlinear cyclic graphs. Unlike differentiable DAG learners that enforce acyclicity through augmented Lagrangian-based solvers, their approach sidesteps these constraints by directly modeling the data likelihood, enabling more efficient and flexible learning of cyclic causal structures. However, their method assumes the absence of unmeasured confounders, which limits its applicability in real-world settings where hidden confounders are often present.

Causal discovery in the presence of latent confounders has seen limited development, with most existing approaches grounded in constraint-based methodologies. Extensions of the PC algorithm, such as the Fast Causal Inference (FCI) algorithm [28], construct a *Partial Ancestral Graph* (PAG) to represent the equivalence class of DAGs in the presence of unmeasured confounders, and can accommodate nonlinear dependencies depending on the chosen conditional independence tests. However, standard FCI does not incorporate interventional data, prompting extensions such as JCI-FCI [29] and related approaches [30] that combine observational and interventional settings. Jaber et al. [31] further advanced this line of work by allowing for unknown interventional targets. Additionally, Forré and Mooij [32] introduced $\sigma$-separation, a generalization of $d$-separation, enabling constraint-based causal discovery in the presence of both cycles and latent confounders. Suzuki and Yang [33] introduce LiNGAM-MMI, a generalization of the ICA-based LiNGAM [34] that quantifies and mitigates confounding via a KL divergence minimization. Integer-programming–based formulations have also been proposed for causal discovery under latent confounding, such as [35, 36]. A few recent approaches, such as Bhattacharya et al. [37], have explored continuous optimization frameworks using differentiable constraints, though these methods are currently limited to linear settings. In parallel, several works exist on causal inference under latent confounding—most notably Abadie et al. [38], Chernozhukov et al. [39]—propose doubly robust estimators that integrate outcome modeling, weighting, and cross-fitting for reliable effect estimation. Overall, a unified framework capable of handling nonlinearity, cycles, latent confounders, and interventions remains largely absent.

**Contributions.** In this work, we tackle three key challenges in causal discovery: *directed cycles*, *nonlinearity*, and *unmeasured confounders*. Our main contributions are:

- We introduce DCCD-CONF, a novel differentiable causal discovery framework for learning nonlinear cyclic relationships under Gaussian exogenous noise, with confounders modeled as correlations in the noise term.

- We show that exact maximization of the proposed score function results in identification of the interventional equivalence class of the ground truth graph.

- We conduct extensive evaluations, comparing DCCD-CONF with state-of-the-art causal discovery methods on both synthetic and real-world datasets.

**Organization.** The paper is structured as follows: Section 2 introduces the problem setup. In Section 3, we present DCCD-CONF, our differentiable framework for nonlinear cyclic causal discovery with unmeasured confounders. We then evaluate its effectiveness on synthetic and real-world datasets in Section 4. Finally, Section 5 concludes the paper.

## 2 Problem Setup

### 2.1 Structural Equations for Cyclic Causal Graphs

Let $\mathcal{G} = (\mathcal{V}, \mathcal{E}, \mathcal{B})$ represent a possibly cyclic *directed mixed graph* (DMG) that encodes the causal dependencies between the variables in the vertex set $\mathcal{V} = [d]$, where $[d] = \{1, \ldots, d\}$. $\mathcal{E}$ denotes the set of directional edges of the form $i \rightarrow j$ in $\mathcal{G}$, and $\mathcal{B}$ denotes the set of bidirectional edges of the form $i \leftrightarrow j$ in $\mathcal{G}$. Each node $i$ is associated with a random variable $X_i$ with the directed edge $i \rightarrow j \in \mathcal{E}$ representing a causal relation between $X_i$ and $X_j$, and the bidirectional edge $i \leftrightarrow j \in \mathcal{B}$ indicates the presence of a hidden confounder between $X_i$ and $X_j$. Following the framework proposed by Bollen [40] and Pearl [41], we use *structural equations model* (SEM) to algebraically describe the system:

$$X_i = F_i(\boldsymbol{X}_{\mathrm{pa}_{\mathcal{G}}(i)}, Z_i), \quad i = 1, \ldots, d, \tag{1}$$

where $\mathrm{pa}_{\mathcal{G}}(i) := \{j \in [d] : j \rightarrow i \in \mathcal{E}\}$ represents the parent set of $X_i$ in $\mathcal{G}$, and $\boldsymbol{X}_{\mathrm{pa}_{\mathcal{G}}(i)}$ denotes the components of $\boldsymbol{X} = (X_1, \ldots, X_d)$ indexed by the parent set $\mathrm{pa}_{\mathcal{G}}(i)$. We exclude *self-loops* (edges of the form $X_i \rightarrow X_i$) from $\mathcal{G}$, as their presence can lead to identifiability challenges [42]. The function $F_i$, referred to as *causal mechanism*, encodes the functional relationship between $X_i$ and its parents $\boldsymbol{X}_{\mathrm{pa}_{\mathcal{G}}(i)}$, and the exogenous noise variable $Z_i$.

The collection of exogenous noise variables $\boldsymbol{Z} = (Z_1, \ldots, Z_d)$ account for the stochastic nature as well as the confounding observed in the system. We make the assumption that the exogenous noise vector follows a Gaussian distributions: $\boldsymbol{Z} \sim \mathcal{N}(\boldsymbol{0}, \boldsymbol{\Sigma}_Z)$. Notably, if $(\boldsymbol{\Sigma}_Z)_{ij} \neq 0$, then variables $X_i$ and $X_j$ are confounded, i.e., $i \leftrightarrow j \in \mathcal{B}$. In other words, confounding is modeled through correlations in the exogenous noise variables. Intuitively, if $X_i$ and $X_j$ share a hidden cause, their unexplained variation (the part not accounted for by their observed parents) will tend to move together. By allowing the noise terms $Z_i$ and $Z_j$ to be correlated, this shared influence can be effectively captured. This formulation generalizes prior work by allowing cycles, extending both nonlinear cyclic models that assume independent noise terms [27], and acyclic models without confounders [43].

By collecting all the causal mechanisms into the joint function $\mathbf{F} = (F_1, \ldots, F_d)$, we can then combine (1) over $i = 1, \ldots, d$ to obtain the equation

$$\boldsymbol{X} = \mathbf{F}(\boldsymbol{X}, \boldsymbol{Z}). \tag{2}$$

We will use (2) to represent the causal system due to its simplicity for subsequent discussion. The observed data represents a snapshot of a dynamical process where the recursive equations in (2) define the system's state at its equilibrium. Thus, in our experiments we assume that the system has reached the equilibrium state. For a given random draw of $\boldsymbol{Z}$, the value of $\boldsymbol{X}$ is defined as the solution to (2). To that end, we assume that (2) admits a unique fixed point for any given $\boldsymbol{Z}$. We refer to the map $\mathbf{f}_x : \boldsymbol{X} \mapsto \boldsymbol{Z}$ as the forward map, and $\mathbf{f}_z : \boldsymbol{Z} \mapsto \boldsymbol{X}$ as the reverse map. In Section 3.1, we show that the chosen parametric family of functions indeed guarantees the existence of a unique fixed point. Under these restrictions, the probability density of $\boldsymbol{X}$ is well defined and is given by

$$p_{\mathcal{G}}(\boldsymbol{X}) = p_Z\big(\mathbf{f}_x(\boldsymbol{X})\big)\big|\det\big(\mathbf{J}_{\mathbf{f}_x}(\boldsymbol{X})\big)\big|, \tag{3}$$

where $\mathbf{J}_{\mathbf{f}_x}(\boldsymbol{X})$ denotes the Jacobian matrix of the function $\mathbf{f}_x$ at $\boldsymbol{X}$.

### 2.2 Interventions

In our work, we consider *surgical interventions* [41], also known as *hard interventions*, where all the incoming edges to the intervened nodes are removed from $\mathcal{G}$. Given a set of intervened upon nodes (also known as *interventional targets*), denoted as $I \subseteq \mathcal{V}$, the structural equations in (1) are modified as follows

$$X_i = \begin{cases} C_i, & \text{if } X_i \in I, \\ F_i(\boldsymbol{X}_{\mathrm{pa}_{\mathcal{G}}(i)}, Z_i), & \text{if } X_i \notin I, \end{cases} \tag{4}$$

where $C_i$ is a random variable sampled from a known distribution, i.e., $C_i \sim p_I(C_i)$. We denote $\mathrm{do}(I)(\mathcal{G})$ to be the mutilated graph under the intervention $I$ (see Figure 1). Note that $X_i$ is no longer confounded if it is intervened on.

We consider a family of $K$ interventional experiments $\mathcal{I} = \{I_k\}_{k \in [K]}$, where $I_k$ represents the interventional targets for the $k$-th experiment. Let $\mathbf{U}_k \in \{0, 1\}^{d \times d}$ denote a diagonal matrix with $(\mathbf{U}_k)_{ii} = 1$ if $i \notin I_k$, and $(\mathbf{U}_k)_{ii} = 0$ if $i \in I_k$. Similar to the observational setting, (4) can be vectorized to obtain the following form

$$\boldsymbol{X} = \mathbf{U}_k \mathbf{F}(\boldsymbol{X}, \boldsymbol{Z}) + \boldsymbol{C}, \tag{5}$$

where $\boldsymbol{C} = (C_1, \ldots, C_d)$ is a vector with $C_i \sim p_I(C_i)$ if $i \in I_k$, and $C_i = 0$ otherwise. For the interventional targets $I_k \in \mathcal{I}$, let $\mathbf{f}_x^{(I_k)}$ denote the forward map. Similar to the observational setting, we make the following assumption on the set of interventions.

**Assumption 1** (Interventional stability). *Let $\mathcal{I} = \{I_k\}_{k \in [K]}$ be a family of interventional targets. For each $I_k \in \mathcal{I}$, the structural equations in (5) admits a unique fixed point given the exogenous noise vector $\boldsymbol{Z}$.*

Thus, the probability distribution of $\boldsymbol{X}$ for the interventional targets $I_k$ is given by

$$p_{\mathrm{do}(I_k)(\mathcal{G})}(\boldsymbol{X}) = p_I(\boldsymbol{C}) p_Z \left( \left[ \mathbf{f}_x^{(I_k)}(\boldsymbol{X}) \right]_{\mathcal{U}_k} \right) \left| \det \left( \mathbf{J}_{\mathbf{f}_x^{(I_k)}}(\boldsymbol{X}) \right) \right|, \tag{6}$$

where $\mathcal{U}_k = \{i : i \in \mathcal{V} \setminus I\}$ denotes the index of purely observed nodes, and $p_Z \left( \left[ \mathbf{f}_x^{(I_k)}(\boldsymbol{X}) \right]_{\mathcal{U}_k} \right)$ is the marginal distribution of the combined vector $Z$, restricted to the components indexed by $\mathcal{U}_k$.

Given a family of interventions $\mathcal{I}$, our goal is to learn the structure of the DMG by maximizing the log-likelihood of the data, in addition to identifying the variables that are being confounded by the unmeasured confounders $\boldsymbol{Z}$. The next section presents our approach to addressing this problem.

## 3 DCCD-CONF: Differentiable Cyclic Causal Discovery with Confounders

In this section, we present our framework for differentiable learning of cyclic causal structures in the presence of unmeasured confounders. We start by modeling the causal mechanisms, then define the score function used for learning, followed by a theorem that validates its correctness. Finally, we outline the algorithm for estimating the model parameters.

### 3.1 Modeling Causal Mechanism

We model the structural equations in (2) using *implicit flows* [44], which define an invertible mapping between $\boldsymbol{x}$ and $\boldsymbol{z}$ by solving the root of a function $\mathbf{G}(\boldsymbol{x}, \boldsymbol{z}) = \mathbf{0}$, where $\mathbf{G} : \mathbb{R}^{2d} \to \mathbb{R}^d$. Specifically, we take $\mathbf{G}(\boldsymbol{x}, \boldsymbol{z}) = \boldsymbol{x} - \mathbf{F}(\boldsymbol{x}, \boldsymbol{z})$. General implicit mappings, however, do not guarantee invertibility or permit efficient computation of the log-determinant required for evaluating (6). To balance expressiveness with tractability, we adopt the structured form proposed by Lu et al. [44] for the causal mechanism:

$$\mathbf{F}(\boldsymbol{x}, \boldsymbol{z}) = -\mathbf{g}_x(\boldsymbol{x}) + \mathbf{g}_z(\boldsymbol{z}) + \boldsymbol{z}, \tag{7}$$

where $\mathbf{g}_x$ and $\mathbf{g}_z$ are restricted to be contractive functions. A function $\mathbf{g} : \mathbb{R}^d \to \mathbb{R}^d$ is contractive if there exists a constant $L < 1$ such that $\|\mathbf{g}(\boldsymbol{x}) - \mathbf{g}(\boldsymbol{y})\| \leq L \|\boldsymbol{x} - \boldsymbol{y}\|$ for all $\boldsymbol{x}, \boldsymbol{y} \in \mathbb{R}^d$. This contractiveness ensures that the associated implicit map is uniquely solvable and invertible (see Theorem 1 in [44]). In other words, contractivity ensures that the process defined by the SEM converges to an equilibrium state.

Under this formulation, the forward map takes the form $\mathbf{f}_x(\boldsymbol{x}) = (\mathbf{id} + \mathbf{g}_z)^{-1} \circ (\mathbf{id} + \mathbf{g}_x)(\boldsymbol{x})$, where $\mathbf{id}$ denotes the identity map. Given $\boldsymbol{x}$ (or $\boldsymbol{z}$), the corresponding value of $\boldsymbol{z}$ (or $\boldsymbol{x}$) can be computed via a root-finding procedure, i.e., $\boldsymbol{z} = \mathrm{RootFind}(\boldsymbol{x} - \mathbf{F}(\boldsymbol{x}, \cdot))$, specifically, we employ a quasi-Newton method (i.e., Broyden's method [45]) to find the root. To capture more complex nonlinear interactions between the observed variables $\boldsymbol{X}$ and latent confounders $\boldsymbol{Z}$, multiple such implicit blocks can be stacked. This is true since $\mathbf{f}_x$ is highly nonlinear and by suitably parameterizing $\mathbf{g}_x$ and $\mathbf{g}_z$ any nonlinear interaction between $\boldsymbol{x}$ and $\boldsymbol{z}$ can be modeled. For simplicity, we focus on a single implicit flow block for subsequent discussion.

We parameterize the functions $\mathbf{g}_x$ and $\mathbf{g}_z$ using neural networks. The adjacency matrix of the causal graph $\mathcal{G}$ is encoded as a binary matrix $\mathbf{M}^{\mathcal{G}} \in \{0, 1\}^{d \times d}$, representing the presence of directed edges and serving as a mask on the inputs to $\mathbf{g}_x$. The diagonal entries of $\mathbf{M}^{\mathcal{G}}$ are explicitly enforced to be zero to prevent self-loops. Similarly, the identity matrix is used to mask the inputs to $\mathbf{g}_z$. Consequently, the causal mechanism is defined as:

$$[\mathbf{F}_{\boldsymbol{\theta}}(\boldsymbol{X}, \boldsymbol{Z})]_i = \big[ -\mathrm{NN}(\mathbf{M}^{\mathcal{G}}_{*,i} \odot \boldsymbol{X} \mid \boldsymbol{\theta}_x) + \mathrm{NN}(Z_i \mid \boldsymbol{\theta}_z) + Z_i \big]_i, \tag{8}$$

where $\mathrm{NN}(\cdot \mid \boldsymbol{\theta})$ denotes a fully connected neural network parameterized by $\boldsymbol{\theta}$, $\odot$ denotes the Hadamard product, and $\mathbf{M}^{\mathcal{G}}_{*,i}$ is the $i$-th column of $\mathbf{M}^{\mathcal{G}}$. The contractivity of $\mathbf{g}_x$ and $\mathbf{g}_z$ can be enforced by rescaling their weights using spectral normalization [46]. Moreover, the contractive nature of the causal mechanism facilitates efficient computation of the score function used for learning causal graphs, as discussed in Section 3.2.

While the contractivity assumption may seem restrictive, it ensures stability and well-posedness in the presence of directed cycles. If the causal graph is known to be acyclic, this assumption can be relaxed (see Appendix C.1).

## 3.2 Score function

Given a family of interventions $\mathcal{I} = \{I_k\}_{k \in [K]}$, we would like to learn the parameters of the structural equation model, i.e., causal graph structure, causal mechanism, and confounder distribution. To that end, similar to prior work [27, 47, 15] in this domain we employ regularized log-likelihood of the observed nodes as the score function to be maximized. That is,

$$\mathcal{S}_{\mathcal{I}}(\mathcal{G}) := \sup_{\boldsymbol{\theta}, \boldsymbol{\Sigma}_Z} \sum_{k=1}^{K} \mathbb{E}_{\boldsymbol{X} \sim p^{(k)}} \log p_{\mathrm{do}(I_k)(\mathcal{G})}(\boldsymbol{X}) - \lambda |\mathcal{G}| \tag{9}$$

where $p^{(k)}$ is the data generating distribution for the $k$-th interventional experiment $I_k$, $\boldsymbol{\Sigma}_Z$ is the parameter (covariance matrix) governing the confounder distribution $p_Z$, $\boldsymbol{\theta} = (\boldsymbol{\theta}_x, \boldsymbol{\theta}_z)$ is the combined causal mechanism parameters, and $|\mathcal{G}|$ denotes a sparsity enforcing regularizer on the edges of $\mathcal{G}$, and $p_{\mathrm{do}(I_k)(\mathcal{G})}(\boldsymbol{X})$ is given by (6).

We now present the main theoretical result of this paper. The following theorem establishes that, under appropriate assumptions, the graph $\hat{\mathcal{G}}$ estimated by maximizing (9) belongs to the same general directed Markov equivalence class (introduced by [42]) as the ground truth graph $\mathcal{G}^*$ for each interventional setting $I_k \in \mathcal{I}$, denoted as $\hat{\mathcal{G}} \equiv_{\mathcal{I}} \mathcal{G}^*$, see Appendix A.1. Due to space constraints we provide the proof sketch below, see Appendix A.3 for complete proof of Theorem 2.

**Theorem 2.** *Let $\mathcal{I} = \{I_k\}_{k=1}^{K}$ be a family of interventional targets, let $\mathcal{G}^*$ denote the ground truth directed mixed graph, let $p^{(k)}$ denote the data generating distribution for $I_k$, and $\hat{\mathcal{G}} := \arg\max_{\mathcal{G}} \mathcal{S}(\mathcal{G})$. Then, under the Assumptions 1, A.13, A.14, and A.15, and for a suitably chosen $\lambda > 0$, we have that $\hat{\mathcal{G}} \equiv_{\mathcal{I}} \mathcal{G}^*$. That is, $\hat{\mathcal{G}}$ is $\mathcal{I}$-Markov equivalent to $\mathcal{G}^*$.*

Theorem 2 rests on three key assumptions. Assumption A.13 ensures that the data-generating distribution lies within the model class, while Assumption A.14 guarantees that every statistical independence in the data corresponds to a $\sigma$-separation in the ground-truth graph. Finally, Assumption 1 prevents the score function from diverging to infinity.

*Proof (Sketch).* Building on the characterization of general directed Markov equivalence class by Bongers et al. [42], extended to the interventional setting, we show that any graph outside this equivalence class has a strictly lower score than the ground truth graph $\mathcal{G}^*$. This follows from the fact that certain independencies present in the data are not captured by graphs outside the equivalence class. Combined with the expressiveness of the model class, this prevents such graphs from fitting the data properly. □

If the intervention set consists of all single-node interventions, $\mathcal{I} = \{I_k\}_{k=1}^{d}$ with $I_k = \{k\}$, Hyttinen et al. [26] showed that the ground truth DMG can be uniquely recovered in the linear setting. Moreover, in the absence of cycles and confounders, this result extends to the nonlinear case, as demonstrated by Brouillard et al. [15]. However, determining the necessary conditions on

interventional targets for perfect recovery in general DMGs with cycles and confounders remains an open problem. Nonetheless, in practice, we find that observational distribution in combination with single-node interventions across all nodes lead to perfect recovery of the ground truth, even in the nonlinear case, as shown in Section 4.

### 3.3 Updating model parameters

In practice, we use gradient based stochastic optimization to maximize (9). For this purpose, following Sethuraman et al. [27] and Brouillard et al. [15], the entries of adjacency matrix $M_{ij}$ are modeled as Bernoulli random variable with parameters $b_{ij}$, grouped into the matrix $\sigma(\mathbf{B})$. We denote $\mathbf{M} \sim \sigma(\mathbf{B})$ to indicate that $M_{ij} \sim \text{Bern}(b_{ij})$ for all $i, j \in [d]$. In this formulation, the sparsity regularizer is $\|\mathbf{M}\|_0$, which is computationally intractable and thus we use the $\ell_1$-norm, $\|\mathbf{M}\|_1$ as a proxy. Consequently, the score function in (9) is replaced by the following relaxation:

$$\hat{\mathcal{S}}_{\mathcal{I}}(\mathbf{B}) := \sup_{\boldsymbol{\theta}, \boldsymbol{\Sigma}_Z} \mathbb{E}_{\mathbf{M} \sim \sigma(\mathbf{B})} \left[ \sum_{k=1}^{K} \sum_{i=1}^{N_k} \log p_{\text{do}(I_k)(\mathcal{G})}(\boldsymbol{x}^{(i,k)}) - \lambda \|\mathbf{M}\|_1 \right], \quad (10)$$

where we replace the expectation with respect to data distribution in (9) with sum over the finite samples, $\boldsymbol{x}^{(i,k)}$ represents the $i$-th data sample in the $k$-the interventional setting. We note that, since $p_Z = \mathcal{N}(\mathbf{0}, \boldsymbol{\Sigma}_Z)$, the covariance of the exogenous confounder vector, $\boldsymbol{\Sigma}_Z$, is implicitly embedded within $p_{\text{do}(I_k)(\mathcal{G})}(\boldsymbol{x}^{(i,k)})$ in the score function.

The optimization of the score function is carried in two steps. First, we optimize $\hat{\mathcal{S}}(\mathbf{B})$ with respect to the neural network parameters $\boldsymbol{\theta}$ and the graph structure parameters $\mathbf{B}$. Next, we optimize $\hat{\mathcal{S}}(\mathbf{B})$ with respect to the parameters of the exogenous noise distribution, $\boldsymbol{\Sigma}_Z$. However, maximizing $\hat{\mathcal{S}}_{\mathcal{I}}(\mathbf{B})$ presents two main challenges: (i) computing $\log p_X(\boldsymbol{X})$ is computationally expensive due to the presence of $|\det(\mathbf{J}_{\mathbf{f}_x^{(I_k)}}(\boldsymbol{X}))|$, which requires $\mathcal{O}(d^2)$ gradient calls, and (ii) updating $\boldsymbol{\Sigma}_Z$ via stochastic gradients could lead to stability issues as $\boldsymbol{\Sigma}_Z$ may loose its positive definiteness.

We now describe how these challenges are addressed, along with the specific procedures for updating the individual model parameters.

#### 3.3.1 Computing log determinant of the Jacobian

As discussed earlier, computing $\log |\mathbf{J}_{\mathbf{f}_x^{(I_k)}}(\boldsymbol{X})|$ is a significant challenge in maximizing the score function $\hat{\mathcal{S}}(\mathcal{B})$. To address this, we utilize the unbiased estimator of the log-determinant of the Jacobian introduced by Behrmann et al. [46], which is based on the power series expansion of $\log(1 + x)$. Since $\mathbf{f}_x^{(I_k)}(\boldsymbol{x}) = (\mathbf{id} + \mathbf{U}_k \mathbf{g}_z)^{-1} \circ (\mathbf{id} + \mathbf{U}_k \mathbf{g}_x)(\boldsymbol{x})$

$$\log \left| \det \left( \mathbf{J}_{\mathbf{f}_x^{(I_k)}}(\boldsymbol{X}) \right) \right| = \log \left| \det \left( \mathbf{I} + \mathbf{J}_{\mathbf{U}_k \mathbf{g}_x}(\boldsymbol{X}) \right) \right| - \log \left| \det \left( \mathbf{I} + \mathbf{J}_{\mathbf{U}_k \mathbf{g}_z}(\boldsymbol{Z}) \right) \right|$$

$$= \sum_{m=1}^{\infty} \frac{(-1)^{m+1}}{m} \left[ \text{Tr} \{ \mathbf{J}_{\mathbf{U}_k \mathbf{g}_x}^m(\boldsymbol{X}) \} - \text{Tr} \{ \mathbf{J}_{\mathbf{U}_k \mathbf{g}_z}^m(\boldsymbol{Z}) \} \right], \quad (11)$$

where $\mathbf{I} \in \mathbb{R}^{d \times d}$ denotes the identity matrix, $\mathbf{J}_{\mathbf{U}_k \mathbf{g}_x}^m$ represents the Jacobian matrix raised to the $m$-th power, and $\text{Tr}$ denotes the trace of matrix. The series in (11) is guaranteed to converge if the causal functions $\mathbf{g}_x$ and $\mathbf{g}_z$ are contractive [48].

In practice, the power series is truncated to a finite number of terms, which may introduce bias into the estimator. To mitigate this issue, we follow the stochastic approach of Chen et al. [49]. Specifically, we sample a random cut-off point $n \sim p_{\mathbb{N}}(n)$ for truncating the power series and weight the $i$-term in the finite series by the inverse probability of the series not ending at $i$. This yields the following unbiased estimator

$$\log \left| \det \left( \mathbf{J}_{\mathbf{f}_x^{(I_k)}}(\boldsymbol{X}) \right) \right| = \mathbb{E}_{n \sim p_{\mathbb{N}}(N)} \left[ \sum_{m=1}^{n} \frac{(-1)^{m+1}}{m} \cdot \frac{\text{Tr} \{ \mathbf{J}_{\mathbf{U}_k \mathbf{g}_x}^m(\boldsymbol{X}) \} - \text{Tr} \{ \mathbf{J}_{\mathbf{U}_k \mathbf{g}_z}^m(\boldsymbol{Z}) \}}{p_{\mathbb{N}}(\ell \geq m)} \right]. \quad (12)$$

The gradient calls can be reduced even further using the *Hutchinson trace estimator* [50], see Appendix B for more details.

### 3.3.2 Updating neural network and graph parameters.

In the first step of the parameter update, keeping $\boldsymbol{\Sigma}_Z$ fixed, the parameters of the neural network $\boldsymbol{\theta}$ and the graph structure $\mathbf{B}$ are updated using the backpropagation algorithm with stochastic gradients. The gradient of the score function $\hat{\mathcal{S}}_{\mathcal{I}}(\mathbf{B})$ with respect to $\mathbf{B}$ is computed using the Straight-Through Gumbel estimator. This involves using Bernoulli samples in the forward pass while computing score, and using samples from Gumbel-Softmax distribution in the backward pass to compute the gradient, which can be differentiated using the reparameterization trick [51].

### 3.3.3 Updating the confounder-noise distribution parameters

In second parameter update step, we fix the value of $\boldsymbol{\theta}$ and $\mathbf{B}$ and focus on the confounder-noise distribution parameter $\boldsymbol{\Sigma}_Z$. First, consider the case where no interventions are applied, i.e, $I_k = \emptyset$. Note that the dependence of $\hat{\mathcal{S}}(\mathbf{B})$ on $\boldsymbol{\Sigma}_Z$ arises solely from $p_Z$, which is embedded within $p_{\text{do}(I_k)(\mathcal{G})}(\boldsymbol{X})$. Therefore, we can thus ignore the remaining terms in $\hat{\mathcal{S}}(\mathbf{B})$ and focus exclusively on $p_Z$. Let $\{\boldsymbol{x}^{(i)}\}_{i=1}^N$ denote the observational data. From the forward map, we have $\boldsymbol{z}^{(i)} = \mathbf{f}_x(\boldsymbol{x}^{(i)})$. Given that $p_Z = \mathcal{N}(\mathbf{0}, \boldsymbol{\Sigma}_Z)$, the relevant parts of $\hat{\mathcal{S}}(\mathbf{B})$ with respect to $\boldsymbol{\Sigma}_Z$, denoted as $\tilde{\mathcal{L}}(I_k)$, are expressed as:

$$\tilde{\mathcal{L}}(I_k) = \sup_{\boldsymbol{\Sigma}_Z} \sum_{i=1}^N -\frac{1}{2}\log|\boldsymbol{\Sigma}_Z| - \frac{(\boldsymbol{z}^{(i)})^\top \boldsymbol{\Sigma}_Z^{-1} \boldsymbol{z}^{(i)}}{2}. \tag{13}$$

Simplifying (13) yields a more convenient form:

$$\tilde{\mathcal{L}}(I_k) = \sup_{\boldsymbol{\Sigma}_Z} -\text{Tr}(\mathbf{S}\boldsymbol{\Sigma}_Z^{-1}) - \log|\boldsymbol{\Sigma}_Z|, \tag{14}$$

where $\mathbf{S} = \frac{1}{n}\sum_{i=1}^N \boldsymbol{z}^{(i)}(\boldsymbol{z}^{(i)})^\top$ is the sample covariance of $\boldsymbol{Z}$.

Maximizing (14) directly using backpropagation and stochastic gradients results in stability issues as $\boldsymbol{\Sigma}_Z$ may lose its positive definiteness. However, Friedman et al. [52] demonstrated that the sparsity-regularized version of (14) is a concave optimization problem in $\boldsymbol{\Sigma}_Z^{-1}$ that can be efficiently solved by optimizing the columns of $\boldsymbol{\Sigma}_Z$ individually. This is achieved by formulating the column recovery as a lasso regression problem. We adopt this strategy while updating the $\boldsymbol{\Sigma}_Z$ during the maximization of $\hat{\mathcal{S}}(\mathbf{B})$.

Let $\mathbf{W} = \boldsymbol{\Sigma}_Z$ be the estimate of the covariance matrix. We reorder $\mathbf{W}$ such that the column and row being updated can be placed at the end, resulting in the following partition

$$\mathbf{W} = \begin{pmatrix} \mathbf{W}_{11} & \boldsymbol{w}_{12} \\ \boldsymbol{w}_{12}^\top & w_{22} \end{pmatrix}, \quad \mathbf{S} = \begin{pmatrix} \mathbf{S}_{11} & \boldsymbol{s}_{12} \\ \boldsymbol{s}_{12}^\top & s_{22} \end{pmatrix}. \tag{15}$$

Then, as shown by Friedman et al. [52], $\boldsymbol{w}_{12} = \mathbf{W}_{11}\boldsymbol{\beta}$, where $\boldsymbol{\beta}$ is the solution to the following lasso regression problem, denoted as $\texttt{lasso}(\mathbf{W}_{11}, \boldsymbol{s}_{12}, \rho)$:

$$\min_{\boldsymbol{\beta}} \frac{1}{2}\|\mathbf{W}_{11}^{1/2}\boldsymbol{\beta} - \boldsymbol{y}\|^2 + \rho\|\boldsymbol{\beta}\|_1, \tag{16}$$

where $\boldsymbol{y} = \mathbf{W}_{11}^{-1/2}\boldsymbol{s}_{12}$, and $\rho$ is the regularization constant that promotes sparsity in $\boldsymbol{\Sigma}_Z^{-1}$.

In an interventional setting $I_k$, the dependence of $\hat{\mathcal{S}}(\mathbf{B})$ on $\boldsymbol{\Sigma}_Z$ arises from the marginal distribution of $\boldsymbol{Z}$ restricted to components indexed by $\mathcal{U}_k$, i.e., purely observed nodes. Since $\boldsymbol{Z}$ follows a Gaussian distribution, $\boldsymbol{Z}_{\mathcal{U}_k}$ also follows a Gaussian distribution with $\boldsymbol{Z}_{\mathcal{U}_k} \sim \mathcal{N}(\mathbf{0}, \hat{\boldsymbol{\Sigma}}_{I_k})$. From the properties of Gaussian distribution [53], we have $\tilde{\boldsymbol{\Sigma}}_{I_k} = (\boldsymbol{\Sigma}_Z)_{\mathcal{U}_k, \mathcal{U}_k}$. Consequently, for the interventional setting $I_k$, (14) becomes

$$\mathcal{L}(I_k) = \sup_{\boldsymbol{\Sigma}_Z} -\text{Tr}\big(\mathbf{S}_{I_k}(\boldsymbol{\Sigma}_Z)_{\mathcal{U}_k, \mathcal{U}_k}^{-1}\big) - \log\big|(\boldsymbol{\Sigma}_Z)_{\mathcal{U}_k, \mathcal{U}_k}\big|, \tag{17}$$

where $\mathbf{S}_{I_k} = \frac{1}{n}\sum_{i=1}^N \boldsymbol{z}_{\mathcal{U}_k}^{(i)}(\boldsymbol{z}_{\mathcal{U}_k}^{(i)})^\top$ is the sample covariance of $\boldsymbol{Z}$ corresponding to the purely observed nodes. In this case, we set $\mathbf{W} = (\boldsymbol{\Sigma}_Z)_{\mathcal{U}_k, \mathcal{U}_k}$ and the rest of the update procedure remains the same. The overall parameter update procedure is summarized in Algorithm 1 in Appendix B.

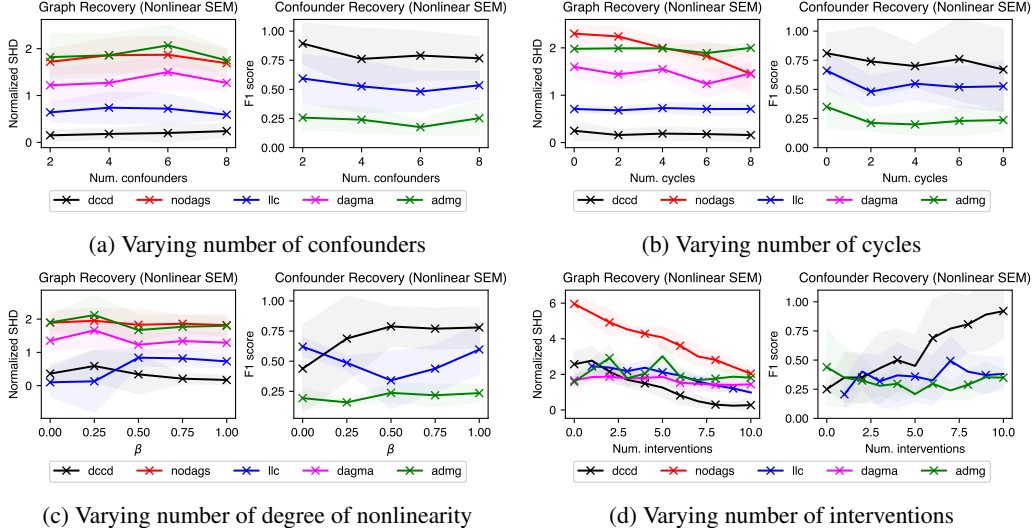

(a) Varying number of confounders

(b) Varying number of cycles

(c) Varying number of degree of nonlinearity

(d) Varying number of interventions

Figure 2: Performance of causal graph and confounder recovery under varying problem dimensions. In all the cases the number of observed variables is fixed at $d = 10$. (Top row, left column) number of latent confounders ranges from 2 to 8, (top row, right column) number of cycles ranges from 0 to 8. (Bottom row, left column) the degree of nonlinearity $\beta$ is varied between 0 and 1, (bottom row, right column) the number of training interventions is varied between 0 and 10.

# 4 Experiments

The code for DCCD-CONF is available at the repository: `https://github.com/muralikgs/dccd_conf`.

We evaluated DCCD-CONF on both synthetic and real-world datasets, comparing its performance against several state-of-the-art baselines: NODAGS-Flow [27], LLC [26], DAGMA [54], and the linear ADMG recovery method proposed by Bhattacharya et al. [37] (which we refer to as ADMG). NODAGS-Flow learns nonlinear cyclic causal graphs but does not model unmeasured confounders. LLC accounts for confounders but is limited to linear cyclic SEMs. DAGMA handles nonlinearity under causal sufficiency while being limited to acyclic graphs. ADMG handles confounding but is limited to acyclic graphs and linear SEMs. Note that both DAGMA and ADMG do not natively support interventional data and hence we use these models in combination with the Joint Causal Inference (JCI) framework [29] and treat interventions as multiple contexts. We also include a comparison between DCCD-CONF and two constraint based models LiNGAM-MMI [33] and JCI-FCI [29] in the Appendix (see Appendix C).

## 4.1 Synthetic data

In all synthetic experiments, the cyclic graphs were generated using Erdős-Rényi (ER) random graph model with the outgoing edge density set to 2. We evaluated DCCD-CONF and the baselines on both linear as well as nonlinear SEMs described in Section 2. Our training data set consists of observational data and single-node interventional over all the nodes in the graph, i.e, $\mathcal{I} = \{\emptyset, \{1\}, \dots, \{d\}\}$ (unless stated otherwise), with $N_k = 500$ samples per intervention. Furthermore, in all the experiments presented here, the SEM was constrained to be contractive. However, we also compare the performance of DCCD-CONF to the baselines on non-contractive SEMs in the appendix. For causal graph recovery (directed edges), we use the *normalized structural Hamming distance* (SHD) as the error metric. SHD counts the number of operations (addition, deletion, and reversal) needed to match the estimated causal graph to the ground truth, and normalization is done with respect to the number of nodes in the graph (lower the better). For confounder identification (bidirectional edges), we compare the non-diagonal entries of the estimated confounder-noise covariance matrix to those of the ground truth. We use F1 score as the error metric (higher the better). More details regarding the experimental setup is provided in Appendix B.

**Impact of confounder count.**    We evaluate the performance of DCCD-CONF and the baselines using the previously defined error metrics, varying the confounder ratio (number of confounders divided by the number of nodes) from 0.2 to 0.8. In this case, the number of nodes in the graph is set to $d = 10$. The results, summarized in Figure 2a, show that DCCD-CONF consistently achieves lower SHD across all confounder ratios in both linear and nonlinear SEMs. Notably, in nonlinear SEMs, DCCD-CONF outperforms all baselines in causal graph recovery. Additionally, it demonstrates competitive results in confounder identification, highlighting its robustness in both tasks.

**Impact of number of cycles.**    With $d = 10$ nodes and a confounder ratio of 0.3, we vary the number of cycles in the graph from 0 to 8. Figure 2b compares the performance of DCCD-CONF with the baselines under this setting. As shown, increasing the number of cycles does not lead to any noticeable degradation in performance for either directed or bidirected edge recovery.

**Impact of degree of nonlinearity.**    In this experiment, we vary the degree of nonlinearity in the SEM by adjusting $\beta$ between 0 and 1, where

$$\boldsymbol{x} = (1 - \beta)(\mathbf{W}^\top \boldsymbol{x} + \boldsymbol{z}) + \beta \tanh(\mathbf{W}^\top \boldsymbol{x} + \boldsymbol{z}).$$

The SEM is fully linear when $\beta = 0$ and fully nonlinear when $\beta = 1$. Figure 2c summarizes the results. As shown, DCCD-CONF attains the highest performance as $\beta$ approaches one, for both directed and bidirected edge recovery. When $\beta$ is small (i.e., the system is more linear), LLC slightly outperforms DCCD-CONF, with both models performing comparably around $\beta = 0.25$.

**Impact of number of interventions.**    In this section, we evaluate graph recovery performance as the number of training interventions $K$ varies from 0 to $d$, with $d = 10$ fixed. The case $K = 0$ corresponds to the observational dataset. Results for the nonlinear SEM setting are presented in Figure 2d. As illustrated, with fewer interventions all DCCD-CONF and the baselines tend to exhibit similar performance (less then 3 interventions). As the numbre of interventions increase, the performance gap widens with DCCD-CONF dominating all of the baselines. It is also worth noting that LLC cannot operate in the purely observational setting ($K = 0$).

**Scaling with nodes.**    We compare the performance of DCCD-CONF and the baselines as the number of nodes ($d$) varies from 10 to 80, with results summarized in Figure 3. The number of confounders is set to $0.3d$. As the number of nodes increases, SHD rises across all methods, reflecting the increased difficulty of causal graph recovery in larger graphs. However, DCCD-CONF consistently outperforms the baselines in many cases, achieving lower SHD and higher F1 score, suggesting superior scalability with increasing graph size.

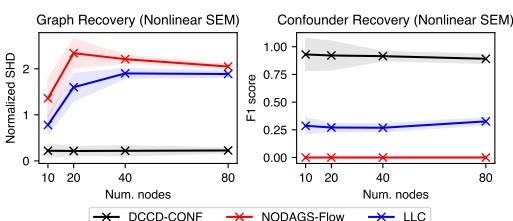

Figure 3: Performance comparison between DCCD-CONF and the baselines and $d$ is varied between 10 and 80.

## 4.2   Real World data

We evaluate DCCD-CONF on learning the causal graph structure of a gene regulatory network from real-world gene expression data with genetic interventions. Specifically, we use the Perturb-CITE-seq dataset [55], which contains gene expression data from 218,331 melanoma cells across three conditions: (i) control, (ii) co-culture, and (iii) IFN-$\gamma$. Due to computational constraints, we restrict our analysis to a subset of 61 genes from the 20,000 genes in the genome, following the experimental setup

Table 1: Results on Perturb-CITE-seq [55] gene perturbation dataset. The table presents the average Negative Log-Likelihood (NLL) on the test set, averaged over multiple trials (standard deviation is reported within parentheses).

| Method | Control | Co-Culture | IFN-$\gamma$ |
|---|---|---|---|
| DCCD-CONF | **1.375** (0.103) | **1.245** (0.039) | **1.235** (0.338) |
| NODAGS | 1.465 (0.015) | 1.406 (0.012) | 1.504 (0.009) |
| LLC | 1.385 (0.039) | 1.325 (0.029) | 1.430 (0.048) |
| DCDI | 1.523 (0.036) | 1.367 (0.018) | 1.517 (0.041) |

of Sethuraman et al. [27] (see Appendix B for details). Each cell condition is treated as a separate dataset consisting of single-node interventions on the selected 61 genes.

Since the dataset does not provide a ground truth causal graph, SHD cannot be used for direct performance comparison. Instead, we assess DCCD-CONF and the baselines based on predictive performance over unseen interventions. To evaluate performance, we split each dataset 90-10, using the smaller portion as the test set, and measure performance using negative log-likelihood (NLL) on the test data after model training (lower the better). The results are presented in Table 1. From Table 1, we can see that DCCD-CONF outperforms all the baselines across all the three cell conditions, showcasing the efficacy of the model and prevalence of confounders in real-world systems. Additionally, we also report the performance of DCCD-CONF and the baselines with respect to MAE on the test data error metric in Table 3 in Appendix C with two additional baselines: DCDFG [47] and Bicycle [56].

**Additional experiments.** Additionally, we also provide results in Appendix C for the following settings: (i) performance comparison on non-contractive SEMs when the underlying graph is restricted to DAGs, (ii) performance comparison as a function of training data size, (iii) performance comparison as a function of noise variance, (iv) performance comparison as a function outgoing edge density, and (v) performance comparison between DCCD-CONF and additional baselines: JCI-FCI and LiNGAM-MMI.

## 5   Discussion

In this work, we introduced DCCD-CONF, a novel differentiable causal discovery framework that handles directed cycles and unmeasured confounders, assuming Gaussian exogenous noise. It models causal mechanisms via neural networks and learns the causal graph structure by maximizing penalized data likelihood. We provide consistency guarantees in the large-sample regime and demonstrate, through extensive synthetic and real-world experiments, that DCCD-CONF outperforms state-of-the-art methods, maintaining robustness with increasing confounders and graph size. On the Perturb-CITE-seq dataset, our model achieves superior predictive accuracy.

While the focus of this work is limited to Gaussian exogenous noise, we plan to investigate other noise distributions for future research. Other future directions include supporting missing data, and relaxing interventional assumptions by incorporating soft interventions and unknown interventional targets.

## Acknowledgment

This material is based upon work supported by the National Science Foundation under Grant No. CCF-2007807 and 2502298.

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

# Appendices

The appendices are structured as follows: Appendix A presents the theoretical foundations of differentiable cyclic causal discovery in the presence of unmeasured confounders, including the proof of Theorem 2, and a characterization of the equivalence class of DMGs that maximize the score function. Appendix B provides implementation details of DCCD-CONF and the baselines. Finally, Appendix C provides additional experimental results comparing DCCD-CONF with the baselines.

## A  Theory

In this section, we establish the theoretical foundations of differentiable cyclic causal discovery in the presence of unmeasured confounders. We begin by reviewing key definitions and results from prior work that are essential for proving Theorem 2, starting with fundamental graph terminology.

### A.1  Preliminaries

Consider a directed mixed graph $\mathcal{G} = (\mathcal{V}, \mathcal{E}, \mathcal{B})$. A *path* $\pi$ between nodes $i$ and $j$ is a sequence $(i_0, \varepsilon_1, i_1, \ldots, \varepsilon_n, i_n)$, where $\{i_0, \ldots, i_n\} \subseteq \mathcal{V}$ and $\{\varepsilon_1, \ldots, \varepsilon_n\} \subseteq \mathcal{E} \cup \mathcal{B}$, with $i_0 = i$ and $i_n = j$. A path is *directed* if each edge $\varepsilon_k$ follows the form $i_{k-1} \to i_k$ for all $k \in [n]$. A cycle through node $i$ consists of a directed path from $i$ to some node $j$ and an additional edge $j \to i$. For any node $i \in \mathcal{V}$, the *ancestor set* is defined as $\mathrm{an}_{\mathcal{G}}(i) := \{j \in \mathcal{V} \mid \text{a directed path from } j \text{ to } i \text{ exists in } \mathcal{G}\}$, while the *descendant set* is given by $\mathrm{de}_{\mathcal{G}}(i) := \{j \in \mathcal{V} \mid \text{a directed path from } i \text{ to } j \text{ exists in } \mathcal{G}\}$. The *spouse set* of a node $i$ is defined as $\mathrm{sp}_{\mathcal{G}}(i) := \{j \in \mathcal{V} \mid j \leftrightarrow i \in \mathcal{B}\}$. If $i$ is both a spouse and an ancestor of $j$, this creates a *almost directed cycle*. A mixed graph is called *ancestral* if it contains neither a directed or an almost directed cycle. The *strongly connected component* of i, denoted $\mathrm{sc}_{\mathcal{G}}(i)$, is the intersection of its ancestors and descendants: $\mathrm{sc}_{\mathcal{G}}(i) = \mathrm{an}_{\mathcal{G}}(i) \cap \mathrm{de}_{\mathcal{G}}(i)$. The *district* of a node $i \in \mathcal{V}$ is defined as $\mathrm{dis}_{\mathcal{G}}(i) = \{j \mid j \leftrightarrow \cdots \leftrightarrow i \in \mathcal{G} \text{ or } i = j\}$. We can apply these definitions to subsets $\mathcal{U} \subseteq \mathcal{V}$ by taking union of over the items of the subset, for instance, $\mathrm{an}_{\mathcal{G}}(\mathcal{U}) = \cup_{i \in \mathcal{U}} \mathrm{an}_{\mathcal{G}}(i)$. A vertex set $\mathcal{A} \subseteq \mathcal{V}$ is said to be *barren* if $i \in \mathcal{A}$ has no descendants in $\mathcal{G}$ that are in $\mathcal{A}$, however, $i$ may have descendants in $\mathcal{G}$ not in $\mathcal{A}$, that is, $\mathrm{barren}_{\mathcal{G}}(\mathcal{A}) = \{i \mid i \in \mathcal{A}; \mathrm{de}_{\mathcal{G}}(i) \cap \mathcal{A} = \{i\}\}$. A subset $\mathcal{A} \subseteq \mathcal{V}$ is *ancestrally closed* if $\mathcal{A}$ contains all of its ancestors. We define $\mathcal{A}(\mathcal{G}) := \{\mathcal{A} \mid \mathrm{an}_{\mathcal{G}}(\mathcal{A}) = \mathcal{A}\}$ as the set of ancestrally closed sets in $\mathcal{G}$.

**Definition A.1** (Collider). *For a directed mixed graph $\mathcal{G} = (\mathcal{V}, \mathcal{E}, \mathcal{B})$, a node $i_k \in \mathcal{V}$ in a path $\pi = (i_0, \varepsilon_1, i_1, \varepsilon_2, \ldots, i_{n-1}, \varepsilon_n, i_n)$ is called a* collider *if $k \neq 0, n$ (non-endpoint) and the two edges $\varepsilon_k, \varepsilon_{k+1}$ have their heads pointed at i, i.e., the subpath $(i_{k-1}, \varepsilon_k, i_k, \varepsilon_{k+1}, i_{k+1})$ is of the form $i_{k-1} \to i_k \leftarrow i_{k+1}, i_{k-1} \leftrightarrow i_k \leftarrow i_{k+1}, i_{k-1} \to i_k \leftrightarrow i_{k+1}, i_{k-1} \leftrightarrow i_k \leftrightarrow i_{k+1}$. The node $i_k$ is called a* non-collider *if $i_k$ is not a collider.*

Note that the end points of a walk are always non-colliders. We now define the notion of $d$-separation extended to DMGs.

**Definition A.2** ($d$-separation). *Let $\mathcal{G} = (\mathcal{V}, \mathcal{E}, \mathcal{B})$ be a directed mixed graph and let $C \subseteq \mathcal{V}$ be a subset of nodes. A path $\pi = (i_0, \varepsilon_1, i_1, \varepsilon_2, \ldots, i_{n-1}, \varepsilon_n, i_n)$ is said to be $d$-blocked given $C$ if*

1. *$\pi$ contains a collider $i_k \notin \mathrm{an}_{\mathcal{G}}(C)$*

2. *$\pi$ contains a non-collider $i_k \in C$.*

*The path $\pi$ is said to be $d$-open given $C$ if it is not $d$-blocked. Two subsets of nodes $A, B \subseteq \mathcal{V}$ is said to be $d$-separated given $C$ if all paths between $a$ and $b$, where $a \in A$ and $b \in B$, is $d$-blocked given $C$, and is denoted by*

$$A \overset{d}{\underset{\mathcal{G}}{\perp}} B \mid C.$$

If the underlying graph is acyclic, $d$-separation implies conditional independence. That is, for subsets of nodes $A, B, C \subseteq \mathcal{V}$,

$$A \overset{d}{\underset{\mathcal{G}}{\perp}} B \mid C \implies \boldsymbol{X}_A \underset{p_{\mathcal{G}}}{\perp} \boldsymbol{X}_B \mid \boldsymbol{X}_C,$$

where $\perp_{p_{\mathcal{G}}}$ denotes conditional independence, and $p_{\mathcal{G}}$ denotes the observational distribution. This is known as the *directed global Markov property* of $\mathcal{G}$ [57]. However, in general, cyclic graphs do

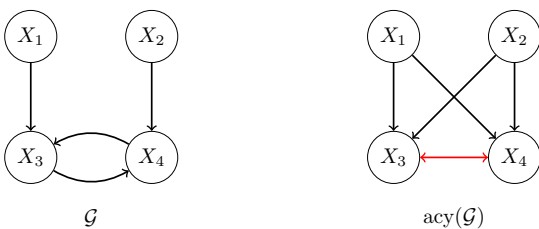

Figure 4: (Left) Illustration of a directed mixed graph that disobeys directed global Markov property. (Right) The graph on the right represents the graph $\mathcal{G}$ after the acyclification process.

not obey the directed global Markov property as shown by the counterexample below taken from [42, 58].

**Example A.3.** *Consider the SEM given by:*

$$f_1(X, Z) = Z_1, \quad f_1(X, Z) = Z_2, \quad f_3(X, Z) = X_1 X_4 + Z_3 \quad f_4(X, Z) = X_2 X_3 + Z_4,$$

*and $p_Z$ is the standard normal distribution. One can check that $X_1$ is not independent of $X_2$ given $\{X_3, X_4\}$. However, the $X_1$ and $X_2$ are d-separated given $\{X_3, X_4\}$ in the graph corresponding to the SEM (see Figure 4).*

Forré and Mooij [57] introduced $\sigma$-separation as a generalization of d-separation to extend the directed global Markov property to cyclic graphs. This concept was motivated by applying $d$-separation to the acyclified version of the DMG. Before delving into $\sigma$-separation, we first define the acyclification procedure of a directed mixed graph, following [42].

**Definition A.4** (Acyclification of a directed mixed graph). *Let $\mathcal{G} = (\mathcal{V}, \mathcal{E}, \mathcal{B})$ denote a directed mixed graph, the acyclification of $\mathcal{G}$ maps $\mathcal{G}$ to the acyclified graph $\mathrm{acy}(\mathcal{G}) = (\mathcal{V}, \hat{\mathcal{E}}, \hat{\mathcal{B}})$, where $j \to i \in \hat{\mathcal{E}}$ if and only if $j \in \mathrm{pa}_{\mathcal{G}}(\mathrm{sc}_{\mathcal{G}}(i)) \setminus \mathrm{sc}_{\mathcal{G}}(i)$, and $i \leftrightarrow j \in \hat{\mathcal{B}}$ if and only if there exists $i' \in \mathrm{sc}_{\mathcal{G}}(i)$ and $j' \in \mathrm{sc}_{\mathcal{G}}(j)$ such that $i' = j'$ or $i' \leftrightarrow j' \in \mathcal{B}$.*

It is important to note that the existence of a acylified graph for an SEM relies on the solvability of the SEM over all the strongly connected components of the DMG corresponding to the SEM. This is to say that we have a solution for $\boldsymbol{X}_{\mathrm{sc}_{\mathcal{G}}(i)}$ given $\boldsymbol{X}_{\mathrm{pa}(\mathrm{sc}_{\mathcal{G}}(i)) \setminus \mathrm{sc}_{\mathcal{G}}(i)}$ and $\boldsymbol{Z}_{\mathrm{sc}_{\mathcal{G}}(i)}$. This is indeed the case as we assume that the forward map $\mathbf{f}_x$ is invertible for the all the SEMs under consideration, see Bongers et al. [42] for more details. Figure 4 illustrates the acyclification process for the graph corresponding to Example A.3.

**Definition A.5** ($\sigma$-separation). *Let $\mathcal{G} = (\mathcal{V}, \mathcal{E}, \mathcal{B})$ be a directed mixed graph and let $C \subseteq \mathcal{V}$ be a subset of nodes. A path $\pi = (i_0, \varepsilon_1, i_1, \varepsilon_2, \ldots, i_{n-1}, \varepsilon_n, i_n)$ is said to be $\sigma$-blocked given $C$ if*

1. *the first node of $\pi$, $i_0 \in C$ or its last node $i_n \in C$, or*

2. *$\pi$ contains a collider $i_k \notin \mathrm{an}_{\mathcal{G}}(C)$*

3. *$\pi$ contains a non-collider $i_k \in C$ that points towards a neighbor that is not in the same strongly connected component as $i_k$ in $\mathcal{G}$, i.e, such that $i_{k-1} \leftarrow i_k$ in $\pi$ and $i_{k-1} \notin \mathrm{sc}_{\mathcal{G}}(i_k)$, or $i_k \to i_{k+1}$ in $\pi$ and $i_{k+1} \notin \mathrm{sc}_{\mathcal{G}}(i_k)$.*

*The path $\pi$ is said to be $\sigma$-open given $C$ if it is not $\sigma$-blocked. Two subsets of nodes $A, B \subseteq \mathcal{V}$ is said to be $\sigma$-separated given $C$ if all paths between $a$ and $b$, where $a \in A$ and $b \in B$, is $\sigma$-blocked given $C$, and is denoted by*

$$A \overset{\sigma}{\underset{\mathcal{G}}{\perp}} B \mid C.$$

Note that $\sigma$-separation reduces to $d$-separation for acyclic graphs, that is, when $\mathrm{sc}_{\mathcal{G}}(i) = \{i\}$ for all $i \in \mathcal{V}$. The following result in [57] relates $\sigma$-separation and $d$-separation.

**Proposition A.6** ([57]). *Let $\mathcal{G} = (\mathcal{V}, \mathcal{E}, \mathcal{B})$ be a directed mixed graph, then for $A, B, C \subseteq \mathcal{V}$,*

$$A \overset{\sigma}{\underset{\mathcal{G}}{\perp}} B \mid C \iff A \overset{d}{\underset{\mathrm{acy}(\mathcal{G})}{\perp}} B \mid C.$$

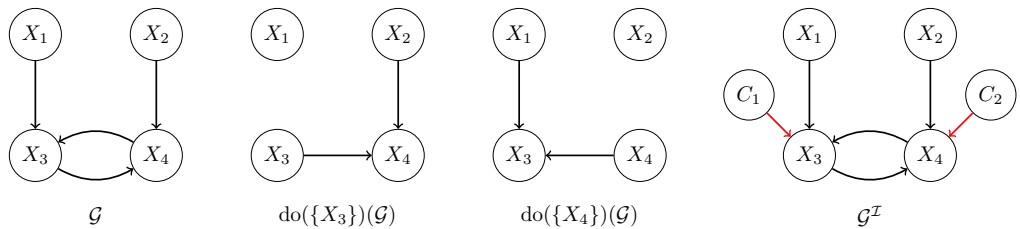

Figure 5: Illustration of the augmented graph $\mathcal{G}^{\mathcal{I}}$ corresponding to the set of interventional targets $\mathcal{I} = \{\emptyset, \{X_3\}, \{X_4\}\}$. $\mathrm{do}(\{X_3\})$ and $\mathrm{do}(\{X_3\})$ corresponds to the graph obtained after hard interventions on $X_3$ and $X_4$ respectively. The augmented graph here is the union of the graphs $\mathcal{G}$, $\mathrm{do}(\{X_3\})$, $\mathrm{do}(\{X_4\})$ along with the context variables.

Using $\sigma$-separation we can now define the general directed global Markov property.

**Definition A.7** (General directed global Markov property [57]). *Let $\mathcal{G} = (\mathcal{V}, \mathcal{E}, \mathcal{B})$ be a directed mixed graph and $p_{\mathcal{G}}$ denote the probability density of the observations $\boldsymbol{X}$. The probability density $p_{\mathcal{G}}$ satisfies the general directed global Markov property if for $A, B, C \subseteq \mathcal{V}$*

$$A \overset{\sigma}{\underset{\mathcal{G}}{\perp}} B \mid C \implies \boldsymbol{X}_A \underset{p_{\mathcal{G}}}{\perp} \boldsymbol{X}_B \mid \boldsymbol{X}_C,$$

*that is, $\boldsymbol{X}_A$ and $\boldsymbol{X}_B$ are conditionally independent given $\boldsymbol{X}_C$.*

### A.2 Joint Causal Modelling and Markov properties

In order to incorporate multiple interventional settings into a single causal modeling framework, we follow the *joint causal model* introduced by [29], where we augment the system with a set of context variables $\boldsymbol{C}^{\mathcal{I}} = (\boldsymbol{C}_1, \dots, \boldsymbol{C}_K)$ each corresponding to a non-empty interventional setting. In this case, $\boldsymbol{C}_k = \emptyset$ for all $k = 1, \dots, K$ corresponds to the observational setting. We construct an augmented graph, denoted by $\mathcal{G}^{\mathcal{I}}$ consisting of both the system variables $\boldsymbol{X}$ and the context variables $\boldsymbol{C}^{\mathcal{I}}$, such that the $\mathrm{ch}_{\mathcal{G}}(\boldsymbol{C}_k) = I_k$, and no context variable has any parent or a spouse. Figure 5 illustrates the augmented graph for the graph from Example A.3 and the intervention sets $\mathcal{I} = \{\emptyset, \{X_3\}, \{X_4\}\}$. The new system containing both the observed variables and the context variables is called the *meta system*. Finally, given a family of interventional targets $\mathcal{I} = \{I_k\}_{k=1}^K$ and the corresponding context variable $\boldsymbol{C}_k$, the structural equations of the meta systems governing the observations $\boldsymbol{X}$ and $\boldsymbol{C}^{\mathcal{I}}$ has the following form:

$$\tilde{F}_i(\boldsymbol{X}_{\mathrm{pa}_{\mathcal{G}}(i)}, \boldsymbol{C}^{\mathcal{I}}_{\mathrm{pa}^{\mathcal{I}}_{\mathcal{G}}(i)}, Z_i) = \begin{cases} (\boldsymbol{C}_k)_i, & \text{if } \exists k \in [K] \text{ s.t. } \boldsymbol{C}_K \neq \emptyset, \text{ and } X_i \in I_k, \\ F_i(\boldsymbol{X}_{\mathrm{pa}_{\mathcal{G}}(i)}, Z_i), & \text{otherwise.} \end{cases}$$

We call the distribution over the context variables $p(\boldsymbol{C}^{\mathcal{I}})$ the *context distributions* and as noted by Mooij et al. [29], the behavior of the system is usually invariant to the context distribution. We assume access to the context distribution as the interventional settings are known apriori. Note that, the observational distribution corresponds to $p_{\mathcal{G}^{\mathcal{I}}}(\boldsymbol{X} \mid C_1 = \cdots = C_K = \emptyset)$. Similarly, the interventional distribution for the interventional setting $I_k$ corresponds to $p_{\mathcal{G}^{\mathcal{I}}}(\boldsymbol{X} \mid \boldsymbol{C}_k = \boldsymbol{\xi}_{I_k}, \boldsymbol{C}_{-k} = \emptyset)$, i.e.,

$$p_{\mathcal{G}^{\mathcal{I}}}(\boldsymbol{X} \mid \boldsymbol{C}_k = \boldsymbol{\xi}_{I_k}, \boldsymbol{C}_{-k} = \emptyset) = p_{\mathrm{do}(I_k)(\mathcal{G})}(\boldsymbol{X})$$

Furthermore,

$$p_{\mathcal{G}^{\mathcal{I}}}(\boldsymbol{C}^{\mathcal{I}}, \boldsymbol{X}) = p_{\mathcal{G}^{\mathcal{I}}}(\boldsymbol{C}^{\mathcal{I}}) p_{\mathcal{G}^{\mathcal{I}}}(\boldsymbol{X} \mid \boldsymbol{C}^{\mathcal{I}}). \tag{18}$$

Recall that for the interventional setting $I_k$, the probability density function governing the observations $\boldsymbol{X}$ is given by (6), which we repeat here for convenience

$$p_{\mathrm{do}(I_k)(\mathcal{G})}(\boldsymbol{X}) = p_I(\boldsymbol{C}) p_Z\left(\left[\mathbf{f}_x^{(I_k)}(\boldsymbol{X})\right]_{\mathcal{U}_k}\right) \left| \det\left(\mathbf{J}_{\mathbf{f}_x^{(I_k)}}(\boldsymbol{X})\right)\right|,$$

**Definition A.8.** *Let $\mathcal{G} = (\mathcal{V}, \mathcal{E}, \mathcal{B})$ be a directed mixed graph, and $\mathcal{I} = \{I_k\}_{k=0}^K$ with $I_0 = \emptyset$ be a family of interventional targets. Let $\mathcal{M}_{\mathcal{I}}(\mathcal{G})$ denote the set of positive densities $p_{\mathcal{G}^{\mathcal{I}}} : \mathbb{R}^{2d} \to \mathbb{R}$ such that $p_{\mathcal{G}^{\mathcal{I}}}$ is given by (18) for all $\mathbf{F} : \mathbb{R}^{2d} \to \mathbb{R}^d$, with $F_i(\boldsymbol{X}, \boldsymbol{Z}) = F_i(\boldsymbol{X}_{\mathrm{pa}_{\mathcal{G}}(i)}, Z_i)$, such that the resulting forward map $\mathbf{f}_x$ is unique and invertible, and $\boldsymbol{\Sigma}_Z \succ 0$ and $(\boldsymbol{\Sigma}_Z)_{ij} \neq 0$ if and only if $i \leftrightarrow j \in \mathcal{B}$.*

**Proposition A.9.** *For a directed mixed graph $\mathcal{G} = (\mathcal{V}, \mathcal{E}, \mathcal{B})$ and a family of interventional targets $\mathcal{I} = \{I_k\}_{k=0}^{K}$ such that $I_0 = \emptyset$, let $p \in \mathcal{M}_{\mathcal{I}}(\mathcal{G})$, then $p$ satisfies the general directed global Markov property relative to $\mathcal{G}^{\mathcal{I}}$.*

*Proof.* For an DMG $\mathcal{G}$ and a choice of $\mathbf{F} : \mathbb{R}^{2d} \to \mathbb{R}^d$ such that $\mathbf{f}_x$ is unique and invertible and $\Sigma \succ 0$, the structural equations are uniquely solvable with respect to each strongly connected component of $\mathcal{G}$. Morover, the addition of context variables in the augmented graph does not introduce any new cycles. Therefore the meta system forms a simple SCM. Thus, from Theorem A.21 in [42], the distribution $p_{\mathcal{G}^{\mathcal{I}}}$ is unique and it satisfies the general directed global Markov property. $\qquad\square$

We now define the notion of interventional Markov equivalence class for DMGs based on the set of distribution induced by them.

**Definition A.10** ($\mathcal{I}$-Markov Equivalence Class)**.** *Two directed mixed graphs $\mathcal{G}_1$ and $\mathcal{G}_2$ are $\mathcal{I}$-Markov equivalent if and only if $\mathcal{M}_{\mathcal{I}}(\mathcal{G}_1) = \mathcal{M}_{\mathcal{I}}(\mathcal{G}_2)$, denoted as $\mathcal{G}_1 \equiv_{\mathcal{I}} \mathcal{G}_2$. The set of all directed mixed graphs that are $\mathcal{I}$-Markov equivalent to $\mathcal{G}_1$ is the $\mathcal{I}$-Markov equivalence class of $\mathcal{G}_1$, denoted as $\mathcal{I}$-MEC($\mathcal{G}_1$).*

From Proposition A.6, for a DMG $\mathcal{G}$ and a family of interventional targets $\mathcal{I} = \{I_k\}_{k=0}^{K}$, any $\sigma$-separation statement in $\mathcal{G}^{\mathcal{I}}$ translates to a $d$-separation statement in the acyclified graph $\text{acy}(\mathcal{G}^{\mathcal{I}})$. Consequently, the acyclified graph $\text{acy}(\mathcal{G}^{\mathcal{I}})$ is equivalent to the augmented graph $\mathcal{G}^{\mathcal{I}}$. Furthermore, by the results of [59], $p_{\mathcal{G}^{\mathcal{I}}}$ admits a factorization, as formalized in the theorem below.

**Theorem A.11** ([59])**.** *A probability distribution $p$ obeys the directed Markov property for an acyclic directed mixed graph $\mathcal{G}$ if and only if for every $A \in \mathcal{A}(\mathcal{G})$,*

$$p(\boldsymbol{X}_A) = \prod_{H \in [A]_{\mathcal{G}}} p(\boldsymbol{X}_H \mid \boldsymbol{X}_{tail(H)}) \tag{19}$$

*where $[A]_{\mathcal{G}}$ denotes a partition of $A$ into sets $\{H_1, \dots, H_k\}$.*

Each term in the factorization above is of the form $p(\boldsymbol{X}_H \mid \boldsymbol{X}_T)$, $H, T \subseteq \mathcal{V}$, and $H \cap T = \emptyset$. Following Richardson [59], Lauritzen and Jensen [60] we refer to $H$ as the *head* of the term $p(\boldsymbol{X}_H \mid \boldsymbol{X}_T)$, and $T$ as the *tail*. An ordered pair of sets $(H, T)$ form the head and tail of the factor associated with $\mathcal{G}$ if and only if all of the following conditions hold:

1. $H = \text{barren}_{\mathcal{G}}(\text{an}_{\mathcal{G}}(H))$,
2. If every nodes $h \in H$ is connected via a path in the graph obtained by removing all the directed edges in the graph $\mathcal{G}$ when restricted to the nodes $\text{an}_{\mathcal{G}}(H)$, and
3. $T = (\text{dis}_{\text{an}_{\mathcal{G}}(H)} \setminus H) \cup \text{pa}_{\mathcal{G}}(\text{dis}_{\text{an}_{\mathcal{G}}(H)})$.

**Proposition A.12.** *Let $\mathcal{G} = (\mathcal{V}, \mathcal{E}, \mathcal{B})$ be a directed mixed graph and $\mathcal{I} = \{I_k\}_{k=0}^{K}$ be a family of interventional targets. The set of interventional distributions $p_{\mathcal{G}^{\mathcal{I}}} \in \mathcal{M}_{\mathcal{I}}(\mathcal{G})$ if and only if $p_{\mathcal{G}^{\mathcal{I}}}$ admits a factorization of the form given by (19).*

*Proof.* Since any $p_{\mathcal{G}^{\mathcal{I}}} \in \mathcal{M}_{\mathcal{I}}(\mathcal{G})$ is also Markov to $\text{acy}(\mathcal{G}^{\mathcal{I}})$, the proposition above is a direct implication of applying of Theorem 19 on $\text{acy}(\mathcal{G}^{\mathcal{I}})$. $\qquad\square$

### A.3 Proof of Theorem 2

We now present the main result of this paper. Recall the score function introduced in Section 3.2,

$$\mathcal{S}_{\mathcal{I}}(\mathcal{G}) := \sup_{\phi} \sum_{k=1}^{K} \mathop{\mathbb{E}}_{\boldsymbol{X} \sim p^{(k)}} \log p_{\text{do}(I_k)(\mathcal{G})}(\boldsymbol{X}) - \lambda |\mathcal{G}|,$$

where $p^{(k)}$ is the data-generating distribution for $I_k \in \mathcal{I}$, and $\phi = \{\boldsymbol{\theta}, \Sigma_Z\}$ represents the set of all model parameters. In the context of the meta system, since we assume access to the context distribution, the score function above is equivalent to the following score:

$$\mathcal{S}_{\mathcal{I}}(\mathcal{G}) := \sup_{\phi} \mathop{\mathbb{E}}_{(\boldsymbol{X}, \boldsymbol{C}) \sim p_{\mathcal{I}}^*} \log p_{\mathcal{G}^{\mathcal{I}}}(\boldsymbol{X}, \boldsymbol{C} \mid \phi) - \lambda |\mathcal{G}|,$$

where $p_{\mathcal{G}^{\mathcal{I}}}(\boldsymbol{X}, \boldsymbol{C} \mid \phi)$ is given by (18) for a specific choice of $\phi$, and $p_{\mathcal{I}}^*$ denotes the joint ground-truth distribution for the observed and the context variables. We define $\mathcal{P}_{\mathcal{I}}(\mathcal{G})$ as the set of all distributions $p_{\mathcal{G}^{\mathcal{I}}}(\boldsymbol{X}, \boldsymbol{C} \mid \phi)$ that can be expressed by the model specified by equations (5) and (8). That is,

$$\mathcal{P}_{\mathcal{I}}(\mathcal{G}) := \{p \mid \exists \phi \text{ s.t } p = p_{\mathcal{G}^{\mathcal{I}}}(\cdot \mid \phi)\}. \tag{20}$$

From the above definition it is clear that $\mathcal{P}_{\mathcal{I}}(\mathcal{G}) \subseteq \mathcal{M}_{\mathcal{I}}(\mathcal{G})$. Theorem 2 relies on the following set of assumptions. The first one ensures that the model is capable of representing the ground truth distribution.

**Assumption A.13** (Sufficient Capacity). *The joint ground truth distribution $p_{\mathcal{I}}^*$ is such that $p_{\mathcal{I}}^* \in \mathcal{P}_{\mathcal{I}}(\mathcal{G}^*)$, where $\mathcal{G}^*$ is the ground truth graph.*

In other words, there exists a $\phi$ such that $p_{\mathcal{I}}^* = p_{\mathcal{G}^{\mathcal{I}}}(\cdot \mid \phi)$. The second assumption generalizes the notion of faithfulness assumption to the interventional setting.

**Assumption A.14** ($\mathcal{I}$-$\sigma$-faithfulness). *Let $\boldsymbol{V} = (\boldsymbol{X}, \boldsymbol{C}^{\mathcal{I}})$, for any subset of nodes $A, B, C \subseteq \mathcal{V} \cup \boldsymbol{C}^{\mathcal{I}}$, and $I_k \in \mathcal{I}$*

$$A \overset{\sigma}{\underset{\mathcal{G}^{\mathcal{I}}}{\not\perp}} B \mid C \implies \boldsymbol{V}_A \underset{p_{\mathcal{G}^{\mathcal{I}}}}{\not\perp} \boldsymbol{V}_B \mid \boldsymbol{V}_C.$$

The above assumption implies that any conditional independency observed in the data must imply a $\sigma$-separation in the corresponding interventional ground truth graph.

**Assumption A.15** (Finite differential entropy). *For $\mathcal{I} = \{I_k\}_{k=0}^K$,*

$$|\mathbb{E}_{p_{\mathcal{I}}^*} \log p_{\mathcal{I}}^*(\boldsymbol{X}, \boldsymbol{C})| < \infty.$$

The above assumption ensures that the hypothetical scenario where $\mathcal{S}(\mathcal{G}^*)$ and $\mathcal{S}(\mathcal{G})$ are both infinity is avoided. This is formalized in the lemma below taken from [15].

**Lemma A.16** (Finiteness of the score function [15]). *Under assumptions A.13 and A.15, $|\mathcal{S}_{\mathcal{I}}(\mathcal{G})| < \infty$.*

From the results of [15], we can now express the difference in score function between $\mathcal{G}^*$ and $\mathcal{G}$ as the minimization of KL diverengence plus the difference in the regularization terms.

**Lemma A.17** (Rewritting the score function [15]). *Under assumptions A.13 and A.15, we have*

$$\mathcal{S}(\mathcal{G}^*) - \mathcal{S}(\mathcal{G}) = \inf_{\phi} D_{KL}(p_{\mathcal{I}}^* \| p_{\mathcal{G}^{\mathcal{I}}}(\cdot \mid \phi)) + \lambda(|\mathcal{G}| - |\mathcal{G}^*|).$$

We will now prove the following technical lemma (adapted from [15]) which we will be used in proving Theorem 2.

**Lemma A.18.** *Let $\mathcal{G} = (\mathcal{V}, \mathcal{E}, \mathcal{B})$ be a directed mixed graph, for a set of interventional targets $\mathcal{I} = \{I_k\}_{k=0}^K$, and $p^* \notin \mathcal{M}_{\mathcal{I}}(\mathcal{G}))$, then*

$$\inf_{p \in \mathcal{M}_{\mathcal{I}}(\mathcal{G}))} D(p^* \| p) > 0.$$

*Proof.* Let $\boldsymbol{V} = (\boldsymbol{X}, \boldsymbol{C}^{\mathcal{I}})$, from theorem A.11, any $p \in \mathcal{M}_{\mathcal{I}}(\mathcal{G}))$ admits a factorization of the form

$$p(\boldsymbol{V}) = \prod_{H \in [\mathcal{V}]_{\mathcal{G}}} p(\boldsymbol{V}_H \mid \boldsymbol{V}_T).$$

Let us define a new distribution $\hat{p}$ as follows:

$$\hat{p}(\boldsymbol{V}) = \prod_{H \in [\mathcal{V}]_{\mathcal{G}}} \hat{p}(\boldsymbol{V}_H \mid \boldsymbol{V}_T),$$

where

$$\hat{p}(\boldsymbol{V}_H \mid \boldsymbol{V}_T) = \frac{p^*(\boldsymbol{V}_H, \boldsymbol{V}_T)}{p^*(\boldsymbol{V}_T)}.$$

From proposition A.12, we see that $\hat{p} \in \mathcal{M}_{\mathcal{I}}(\mathcal{G}))$ and hence $p \neq \hat{p}$. We will show that

$$\hat{p} = \arg \min_{p \in \mathcal{M}_{\mathcal{I}}(\mathcal{G}))} D_{KL}(p^* \| p).$$

For an arbitrary $p \in \mathcal{M}_{\mathcal{I}}(\mathcal{G}))$, consider the following:

$$\mathbb{E}_{p^*} \log \frac{\hat{p}(\boldsymbol{V})}{p(\boldsymbol{V})} = \mathbb{E}_{p^*} \sum_{H \in [\mathcal{V}]_{\mathcal{G}}} \log \frac{p^*(\boldsymbol{V}_H \mid \boldsymbol{V}_T)}{p(\boldsymbol{V}_H \mid \boldsymbol{V}_T)} \tag{21}$$

$$= \sum_{H \in [\mathcal{V}]_{\mathcal{G}}} \mathbb{E}_{p^*} \log \frac{p^*(\boldsymbol{V}_H \mid \boldsymbol{V}_T)}{p(\boldsymbol{V}_H \mid \boldsymbol{V}_T)}. \tag{22}$$

In the equation above, we leverage the linearity of expectation, which holds under Assumption A.15, ensuring that we don't sum infinities of opposite signs. We now show that each term in the right hand side of the above equality is an expectation of KL divergence which is always in $[0, \infty)$.

$$\mathbb{E}_{p^*} \log \frac{p^*(\boldsymbol{V}_H \mid \boldsymbol{V}_T)}{p(\boldsymbol{V}_H \mid \boldsymbol{V}_T)} = \int p^*(\boldsymbol{V}_T) \int p^*(\boldsymbol{V}_H \mid \boldsymbol{V}_T) \log \frac{p^*(\boldsymbol{V}_H \mid \boldsymbol{V}_T)}{p(\boldsymbol{V}_H \mid \boldsymbol{V}_T)} d\boldsymbol{V}_H d\boldsymbol{V}_T \tag{23}$$

$$= \int p^*(\boldsymbol{V}_T) D_{KL}\left(p^*(\cdot \mid \boldsymbol{V}_T) \| p(\cdot \| \boldsymbol{V}_T)\right) d\boldsymbol{V}_T. \tag{24}$$

Thus, $\mathbb{E}_{p^*} \log \frac{\hat{p}(\boldsymbol{V})}{p(\boldsymbol{V})} \in [0, \infty)$.

We now show that $\hat{p} = \arg \min_{p \in \mathcal{M}(\mathrm{do}(I_k)(\mathcal{G}))} D_{KL}(p^* \| p)$:

$$D_{KL}(p^* \| p) = \mathbb{E}_{p^*} \log \frac{p^*(\boldsymbol{V})}{\hat{p}(\boldsymbol{V})} \frac{\hat{p}(\boldsymbol{V})}{p(\boldsymbol{V})} \tag{25}$$

$$= \mathbb{E}_{p^*} \log \frac{p^*(\boldsymbol{V})}{\hat{p}(\boldsymbol{V})} + \mathbb{E}_{p^*} \log \frac{\hat{p}(\boldsymbol{V})}{p(\boldsymbol{V})} \tag{26}$$

$$= D_{KL}(p^* \| \hat{p}) + \mathbb{E}_{p^*} \log \frac{\hat{p}(\boldsymbol{V})}{p(\boldsymbol{V})} \tag{27}$$

$$\geq D_{KL}(p^* \| \hat{p}) > 0. \tag{28}$$

Since the expectations in (26) are both in $[0, \infty)$, splitting the expectation is valid. The very last inequality holds since $p^* \neq \hat{p}$. Thus,

$$\inf_{p \in \mathcal{M}_{\mathcal{I}}(\mathcal{G}))} D(p^* \| p) \geq D_{KL}(p^* \| \hat{p}) > 0.$$

This proves the lemma. $\qquad \square$

We are now ready to prove Theorem 2. Recall,

**Theorem 2.** *Let $\mathcal{I} = \{I_k\}_{k=0}^{K}$ be a family of interventional targets, let $\mathcal{G}^*$ denote the ground truth directed mixed graph, $p^{(k)}$ denote the data generating distribution for $I_k$, and $\hat{\mathcal{G}} := \arg \max_{\mathcal{G}} \mathcal{S}(\mathcal{G})$. Then, under the Assumptions 1, A.13, A.14, and A.15, and for a suitably chosen $\lambda > 0$, we have that $\hat{\mathcal{G}} \equiv_{\mathcal{I}} \mathcal{G}^*$. That is, $\hat{\mathcal{G}}$ is $\mathcal{I}$-Markov equivalent to $\mathcal{G}^*$.*

*Proof.* It is sufficient to show that for $\mathcal{G} \notin \mathcal{I}\text{-MEC}(\mathcal{G}^*)$, the score function of $\hat{\mathcal{G}}$ is strictly lower than the score function of $\mathcal{G}^*$, i.e., $\mathcal{S}(\mathcal{G}^*) > \mathcal{S}(\mathcal{G})$. Since $\mathcal{G} \notin \mathcal{I}\text{-MEC}(\mathcal{G}^*)$ and $p_{\mathcal{I}}^* \in \mathcal{M}_{\mathcal{I}}(\mathcal{G}^*)$ (by Assumption A.13), there must exist subsets of nodes $A, B, C \subseteq \mathcal{V} \cup \boldsymbol{C}^{\mathcal{I}}$ such that either:

$$A \underset{\mathcal{G}}{\overset{\sigma}{\perp}} B \mid C \quad \text{and} \quad A \underset{\mathcal{G}^*}{\overset{\sigma}{\not\perp}} B \mid C,$$

or

$$A \underset{\mathcal{G}}{\overset{\sigma}{\not\perp}} B \mid C \quad \text{and} \quad A \underset{\mathcal{G}^*}{\overset{\sigma}{\perp}} B \mid C,$$

If no such subsets exist, then $\mathcal{G}$ and $\mathcal{G}^*$ impose the same $\sigma$-separation constraints and thus induce the same set of distributions. This would imply that $\mathcal{G} \in \mathcal{I}\text{-MEC}(\mathcal{G}^*)$, contradicting our assumption.

Since $p^*_\mathcal{I} \in \mathcal{M}_\mathcal{I}(\mathcal{G}^*))$, it must be true that $\boldsymbol{V}_A \not\perp_{p^{(k)}} \boldsymbol{V}_B \mid \boldsymbol{V}_C$ (Assumption A.14). Therefore $p\mathcal{I}^*$ doesn't satisfy the general directed Markov property with respect to $\mathcal{G}^\mathcal{I}$ and hence $p^*_\mathcal{I} \notin \mathcal{M}_\mathcal{I}(\mathcal{G})$.

For convenience, let

$$\eta(\mathcal{G}) := \inf_\phi D_{KL}(p^*_\mathcal{I} \| p_{\mathcal{G}^\mathcal{I}}(\cdot \mid \boldsymbol{\phi})).$$

Note that

$$\eta(\mathcal{G}) = \inf_\phi D_{KL}(p^*_\mathcal{I} \| p_{\mathcal{G}^\mathcal{I}}(\cdot \mid \boldsymbol{\phi})) \geq \inf_{p \in \mathcal{M}_\mathcal{I}(\mathcal{G})} D_{KL}(p^{(k)} \| p) > 0,$$

where we use Lemma A.18 for the final inequality. Thus, from Lemma A.17

$$\mathcal{S}(\mathcal{G}^*) - \mathcal{S}(\mathcal{G}) = \eta(\mathcal{G}) + \lambda(|\mathcal{G}| - |\mathcal{G}^*|) \tag{29}$$

Following [15], we now show that by choosing $\lambda$ sufficiently small, the above equation is stictly positive. Note that if $|\mathcal{G}| \geq |\mathcal{G}^*|$ then $\mathcal{S}(\mathcal{G}^*) - \mathcal{S}(\mathcal{G}) > 0$. Let $\mathbb{G}^+ := \{\mathcal{G} \mid |\mathcal{G}| < |\mathcal{G}^*|\}$. Choosing $\lambda$ such that $0 < \lambda < \min_{\mathcal{G} \in \mathbb{G}^+} \frac{\eta(\mathcal{G})}{|\mathcal{G}^*| - |\mathcal{G}|}$ we see that:

$$\lambda < \min_{\mathcal{G} \in \mathbb{G}^+} \frac{\eta(\mathcal{G})}{|\mathcal{G}^*| - |\mathcal{G}|} \tag{30}$$

$$\iff \lambda < \frac{\eta(\mathcal{G})}{|\mathcal{G}^*| - |\mathcal{G}|} \quad \forall \mathcal{G} \in \mathbb{G}^+ \tag{31}$$

$$\iff \lambda(|\mathcal{G}^*| - |\mathcal{G}|) < \eta(\mathcal{G}) \quad \forall \mathcal{G} \in \mathbb{G}^+ \tag{32}$$

$$\iff 0 < \eta(\mathcal{G}) + \lambda(|\mathcal{G}| - |\mathcal{G}^*|) = \mathcal{S}(\mathcal{G}^*) - \mathcal{S}(\mathcal{G}) \quad \forall \mathcal{G} \in \mathbb{G}^+. \tag{33}$$

Thus, every graph outside of the general directed Markov equivalence class of $(\mathcal{G}^*)^\mathcal{I}$ has a strictly lower score. □

### A.4 Characterization of Equivalence Class

Let $\mathcal{G} = (\mathcal{V}, \mathcal{E}, \mathcal{B})$ be a directed mixed graph, and consider a family of interventional targets $\mathcal{I} = I_{k\,k=0}^K$ with $I_0 = \emptyset$. From Proposition A.6, for any graph $\mathcal{G}_1 \in \mathcal{I}\text{-MEC}(\mathcal{G})$, the corresponding augmented graph $\mathcal{G}_1^\mathcal{I}$ is equivalent to the acyclification $\mathrm{acy}(\mathcal{G}^\mathcal{I})$ of $\mathcal{G}^\mathcal{I}$. Several prior works have studied the characterization of equivalence classes of acyclic directed mixed graphs (ADMGs), including [61, 30, 62]. We now provide a graphical notion of the $\mathcal{I}$-Markov equivalence class of a DMG $\mathcal{G}$. A graph $\mathcal{G}$ is said to be *maximal* if there exists no inducing path (relative to the empty set) between any two non-adjacent nodes. An *inducing path* relative to a subset $L$ is a path on which every non-endpoint node $i \notin L$ is a collider on the path and every collider is an ancestor of an endpoint of the path. A *Maximal Ancestral Graph* (MAG) is one that is both ancestral and maximal. Given an ADMG $\mathrm{acy}(\mathcal{G}^\mathcal{I})$, it is possible to construct a MAG over the variable set $\boldsymbol{V} = (\boldsymbol{X}, \boldsymbol{C}^\mathcal{I})$ that preserves both the independence structure and ancestral relationships encoded in $\mathrm{acy}(\mathcal{G}^\mathcal{I})$; see [63] for details. We denote $MAG(\mathrm{acy}(\mathcal{G}^\mathcal{I}))$ to mean MAG that is constructed from $\mathrm{acy}(\mathcal{G}^\mathcal{I})$. Therefore all the independencies encoded in $\mathrm{acy}(\mathcal{G}^\mathcal{I})$ is also present in $MAG(\mathrm{acy}(\mathcal{G}^\mathcal{I}))$. Before present the condition for MAG equivalence, we introduce the notion of discriminating path. A path $\pi = (i_0, \varepsilon_1, \ldots, i_{n-1}, \varepsilon_n, i_n)$ in $\mathrm{acy}(\mathcal{G}^\mathcal{I})$ is called a *discriminating path* for $i_{n-1}$ if (1) $\pi$ includes at lest three edges; (2) $i_{n-1}$ is a non-endpoint node on $\pi$, and is adjacent to $i_n$ on $\pi$; and (3) $i_0$ and $i_n$ are not adjacent, and every node in between $i_0$ and $i_{n-1}$ is a collider on $\pi$ and is a parent of $i_n$. The following theorem from Spirtes and Richardson [61] characterizes the equivalence of MAGs.

**Theorem A.19** (Spirtes and Richardson [61]). *Two MAGs $\mathcal{G}_1$ and $\mathcal{G}_2$ are Markov equivalent if and only if:*

1. *$\mathcal{G}_1$ and $\mathcal{G}_2$ have the same skeleton;*

2. *$\mathcal{G}_1$ and $\mathcal{G}_2$ have the same unshielded colliders; and*

3. *if $\pi$ forms a discriminating path for $i$ in $\mathcal{G}_1$ and $\mathcal{G}_2$, then $i$ is a collider on $\pi$ if and only it is a collider on $\pi$ in $\mathcal{G}_2$.*

Therefore, from Proposition A.6 and Theorem A.19, two DMGs $\mathcal{G}_1$ and $\mathcal{G}_2$ are equivalent if and only if $MAG(\mathrm{acy}(\mathcal{G}_1^\mathcal{I}))$ and $MAG(\mathrm{acy}(\mathcal{G}_2^\mathcal{I}))$ satisfying the conditions of Theorem 2, i.e., $MAG(\mathrm{acy}(\mathcal{G}_1^\mathcal{I}))$ and $MAG(\mathrm{acy}(\mathcal{G}_2^\mathcal{I}))$: (i) have the same skeleton, (ii) same unshielded colliders, and (iii) same discriminating paths with consistent colliders.

# B  Implementation Details

In this section we provide the implementation details of DCCD-CONF and the baseline models along with the details of the experimental setup.

## B.1  Hutchinson trace estimator for computing log determinant of the Jacobian

Computing the log determinant of the Jacobian matrix present in (6) poses a significant challenge. However, following Behrmann et al. [46], in Section 3.3.1 showed that the log-determinant of the Jacobian can be estimated using the following estimator

$$\log \big| \det \big( \mathbf{J}_{\mathbf{f}_x^{(I_k)}}(\boldsymbol{X}) \big) \big| = \mathop{\mathbb{E}}_{n \sim p_{\mathbb{N}}(N)} \left[ \sum_{m=1}^{n} \frac{(-1)^{m+1}}{m} \cdot \frac{\mathrm{Tr}\big\{ \mathbf{J}_{\mathbf{U}_k \mathbf{g}_x}^m(\boldsymbol{X}) \big\} - \mathrm{Tr}\big\{ \mathbf{J}_{\mathbf{U}_k \mathbf{g}_z}^m(\boldsymbol{Z}) \big\}}{p_{\mathbb{N}}(\ell \geq m)} \right].$$

The estimator above still has a major drawback: computing the $\mathrm{Tr}(\mathbf{J}_{\mathbf{U}_k \mathbf{g}})$ still requires $\mathcal{O}(d^2)$ gradient calls to compute exactly. Fortunately, *Hutchinson trace estimator* [50] can be used to stochastically approximate the trace of the Jacobian matrix. This then results in the following estimator that can be computed efficiently via reverse-mode automatic differentiation

$$\log \big| \det \big( \mathbf{J}_{\mathbf{f}_x^{(I_k)}}(\boldsymbol{X}) \big) \big| = \mathop{\mathbb{E}}_{n \sim p_{\mathbb{N}}(N), \boldsymbol{V} \sim \mathcal{N}(\mathbf{0}, \mathbf{I})} \left[ \sum_{m=1}^{n} \frac{(-1)^{m+1}}{m} \right.$$
$$\left. \times \frac{\boldsymbol{V}^\top \big\{ \mathbf{J}_{\mathbf{U}_k \mathbf{g}_x}^m(\boldsymbol{X}) \big\} \boldsymbol{V} - \boldsymbol{V}^\top \big\{ \mathbf{J}_{\mathbf{U}_k \mathbf{g}_z}^m(\boldsymbol{Z}) \big\} \boldsymbol{V}}{p_{\mathbb{N}}(\ell \geq m)} \right]. \quad (34)$$

## B.2  Parameter update via score maximization

As described in Section 3, the model parameters are updated in two stages. In the first stage, the parameters of the neural networks and the Gumbel-Softmax distribution, used to sample adjacency matrices, are updated via backpropagation using stochastic gradient descent. Since (5) forms an implicit block of an implicit normalizing flow, following [44], we directly estimate the gradients of $\hat{\mathcal{S}}(\mathbf{B})$ with respect to $\boldsymbol{x}$ and $\boldsymbol{\phi} = (\boldsymbol{\theta}, \mathbf{B})$. The gradient computation involves two terms: $\frac{\partial}{\partial(\cdot)} \log \det(\mathbf{I} + \mathbf{J}_{\mathbf{g}_x}(\boldsymbol{x}, \boldsymbol{\phi}))$ and $\frac{\partial \hat{\mathcal{S}}}{\partial \boldsymbol{z}} \frac{\partial \boldsymbol{z}}{\partial(\cdot)}$, where $(\cdot)$ is a placeholder for $\boldsymbol{x}$ and $\boldsymbol{\phi}$. From [44, 46], we use the following unbiased estimators for the gradients:

$$\frac{\partial \log \det(\mathbf{I} + \mathbf{J}_{\mathbf{g}_x}(\boldsymbol{x}, \boldsymbol{\phi}))}{\partial(\cdot)} = \mathop{\mathbb{E}}_{n \sim p(N), \boldsymbol{V} \sim \mathcal{N}(\mathbf{0}, \mathbf{I})} \left[ \left( \sum_{k=0}^{n} \frac{(-1)^k}{P(N \geq k)} \boldsymbol{V}^\top \mathbf{J}^k \right) \frac{\partial \mathbf{J}_{\mathbf{g}_x}(\boldsymbol{x}, \boldsymbol{\phi})}{\partial(\cdot)} \boldsymbol{V} \right]. \quad (35)$$

On the other hand, $\frac{\partial \hat{\mathcal{S}}(\mathbf{B})}{\partial \boldsymbol{z}} \frac{\partial \boldsymbol{z}}{\partial(\cdot)}$ can be computed according to the *implicit function theorem* as follows:

$$\frac{\partial \hat{\mathcal{S}}(\mathbf{B})}{\partial \boldsymbol{z}} \frac{\partial \boldsymbol{z}}{\partial(\cdot)} = \frac{\hat{\mathcal{S}}(\mathbf{B})}{\partial \boldsymbol{z}} \mathbf{J}_{\mathbf{G}_z}^{-1}(\boldsymbol{z}) \frac{\mathbf{G}(\boldsymbol{x}, \boldsymbol{z}, \boldsymbol{\phi})}{\partial(\cdot)}, \quad (36)$$

where $\mathbf{G}_z(\boldsymbol{z}) = \mathbf{g}_z(\boldsymbol{z}, \boldsymbol{\phi}) + \boldsymbol{z}$, and recall that $\mathbf{G}(\boldsymbol{x}, \boldsymbol{z}, \boldsymbol{\phi}) = \mathbf{g}_x(\boldsymbol{x}, \boldsymbol{\phi}) + \boldsymbol{x} + \mathbf{g}_z(\boldsymbol{z}, \boldsymbol{\phi}) + \boldsymbol{z}$. See [44] for more details. The procedure SGUPDATE shown in Algorithm 1 performs the gradient computation in (35) and (36).

In the second stage, the entries of the covariance matrix of the endogenous noise distribution are updated column-wise by solving a sequence of Lasso optimization problems. The complete parameter update procedure is summarized in Algorithm 1.

## B.3  DCCD-CONF and the baselines code details

**DCCD-CONF.**  We implemented our framework using the libraries `Pytorch` and `Scikit-learn` in Python and the code used in running the experiments can be found in the following Github repository: `https://github.com/muralikgs/dccd_conf`.

---

**Algorithm 1** PARAMETER UPDATE

---

**Require:** Family of interventional targets $\mathcal{I} = \{I_k\}_{k=1}^{K}$, interventional dataset $\{x^{(i,k)}\}_{i=1,k=1}^{N_k,K}$, regularization coefficients $\lambda$ and $\rho$.

**Ensure:** Learned neural network parameters $\hat{\theta}$, graph structure parameters $\hat{\mathcal{B}}$, confounder-noise distribution parameters $\hat{\Sigma}_Z$.

1: Initialize the parameters: $\theta^{(0)} \sim p_\theta(\theta)$, $\mathcal{B}^{(0)} \sim p_\mathcal{B}(\mathcal{B})$, and $\Sigma_Z = \mathbf{I}$
2: Iteration counter: $t = 0$
3: **while** NOT CONVERGED **do**
4:     **for** $k = 1$ to $K$ **do**
5:         $t \leftarrow t + 1$
6:         $\mathbf{W} \leftarrow (\Sigma_Z^{(t)})_{\mathcal{U}_k, \mathcal{U}_k}$
7:         Compute score function $\tilde{\mathcal{L}}(\mathcal{B}^{(t)}, \theta^{(t)}, \mathbf{W}, I_k)$
8:         $\mathcal{B}^{(t+1)}, \theta^{(t+1)} \leftarrow$ SGUPDATE$(\tilde{\mathcal{L}}, \mathcal{B}^{(t)}, \theta^{(t)})$
9:         **for** $j = 1$ to $d$ **do**
10:            Push $j$-th row and column in $\mathbf{W}$ to the end
11:            $\beta \leftarrow$ lasso$(\mathbf{W}_{11}, s_{12}, \rho)$
12:            $w_{12} \leftarrow \mathbf{W}_{11}\beta$
13:         **end for**
14:         $(\Sigma_Z^{(t+1)})_{\mathcal{U}_k, \mathcal{U}_k} \leftarrow \mathbf{W}$
15:     **end for**
16: **end while**
17: $\hat{\theta}, \hat{\mathcal{B}}, \hat{\Sigma}_Z \leftarrow \theta^{(t)}, \mathcal{B}^{(t)}, \Sigma_Z^{(t)}$
      **return** $\hat{\theta}, \hat{\mathcal{B}}, \hat{\Sigma}_Z$

---

Starting with an initialization of the model parameters $(\theta^{(0)}, \mathbf{B}^{(0)}, \Sigma_Z^{(0)})$, we iteratively alternate between maximizing the score function with respect to $(\theta^{(t)}, \mathbf{B}^{(t)})$ and $\Sigma_Z^{(t)}$, as described in Algorithm 1. Standard stochastic gradient updates are used for $(\theta^{(t)}, \mathbf{B}^{(t)})$, while coordinate gradient descent, implemented via the `Scikit-learn` library, is applied to $\Sigma_Z^{(t)}$. For modeling the causal function $\mathbf{g}_x$, we follow the setup of Sethuraman et al. [27], employing neural networks (NNs) with dependency masks parameterized by a Gumbel-softmax distribution. The log-determinant of the Jacobian is computed using a power series expansion combined with the Hutchinson trace estimator. To mitigate bias from truncating the power series expansion, the number of terms is sampled from a Poisson distribution, as detailed in Section 3.3 and Appendix B.1. The final objective is optimized using the Adam optimizer [64].

The learning rate in all our experiments was set to $10^{-2}$. The neural network models used in our experiments contained one multi-layer perceptron layer. No nonlinearities were added to the neural networks for the linear SEM experiments. We used `tanh` activation for the nonlinear SEM experiments and for the experiments on the perturb-CITE-seq data set. The graph sparsity regularization constant $\lambda$ was set to $10^{-2}$ for all the experiments. The sparsity inducing regularization constant for the inverse covariance matrix of the confounder distribution, $\rho$, was set to $10^{-1}$ in all the experiments. The models were trained and evaluated on NVIDIA RTX6000 GPUs.

**Baselines.** For NODAGS-Flow, we used the code provided by authors [27] available at `https://github.com/Genentech/nodags-flows`. The default values were set for the hyperparameters. We implemented the LLC algorithm based on the details provided in [26]. The implementation can be found within the `codes/baselines` folder in the supplementary materials. For FCI, we used the implementation that is available in the `causallearn` python library (`https://github.com/py-why/causal-learn`). For DCDI, we used the codebase provided by the authors [15], available at `https://github.com/slachapelle/dcdi`. The default hyperparameters were used while training and evaluating the model. For DAGMA and ADMG, we used the codebase provided by the authors [54] and [37], available at `https://github.com/kevinsbello/dagma` and `https://gitlab.com/rbhatta8/dcd` respectively. We implemented LiNGAM-MMI based on the details provided by the authors [33].

## B.4 Experimental setup

In this section, we describe how the data sets were generated for the various experiments conducted.

### B.4.1 Synthetic Experiments

We begin by sampling a directed graph using the Erdős-Rényi (ER) random graph model with an edge density of 2 unless specified otherwise, which determines the directed edges in the DMG $\mathcal{G}$. Next, we generate a random matrix and project it onto the space of positive definite matrices to obtain the confounder covariance matrix $\mathbf{\Sigma}_Z$, setting the maximum exogenous noise standard deviation to 0.5 unless specified otherwise. The nonzero off-diagonal entries of $\mathbf{\Sigma}_Z$ correspond to the bidirectional edges in $\mathcal{G}$. For the linear SEM, edge weights are sampled uniformly from $\text{Unif}((-0.9, -0.2) \cup (0.2, 0.9))$. In all experiments except those on non-contractive SEMs, the edge weight matrix is rescaled to ensure a Lipschitz constant of less than one. For nonlinear SEMs, we apply a `tanh` nonlinearity to the linear system defined by the edge weights, i.e.,

$$\boldsymbol{x} = \tanh(\mathbf{W}^\top \boldsymbol{x} + \boldsymbol{z}),$$

where $\mathbf{W}$ is the weighted adjacency matrix. In all the experiments, the training data consisted of 500 samples per interventional setting (unless specified otherwise).

**Impact of Confounder Count**   In this experiment, the number of observed nodes in the graph is fixed at d = 10 . Training data consists of combination of observational data and single-node interventions over all nodes, i.e., $\mathcal{I} = \emptyset \cup \{\{i\} \mid i \in [d]\}$. The confounder ratio (number of confounders divided by the number of nodes) is varied from 0.2 to 0.8.

**Impact of Cycles**   We fix the number of nodes in the graph $d = 10$, The number of cycles in the graph is varied between 0 and 8, with the confounder ratio set to 0.3. The training data consists of observational data as well as single node interventions over all the nodes in the graph.

**Impact of Nonlinearity**   In this setting, the degree of nonlinearity (controlled by $\beta$) is varied between 0 and 1. That is, the data is generated from the following SEM:

$$\boldsymbol{x} = (1 - \beta)(\mathbf{W}^\top \boldsymbol{x} + \boldsymbol{z}) + \beta \tanh(\mathbf{W}^\top \boldsymbol{x} + \boldsymbol{z}).$$

Here, $\beta = 0$ implies the SEM is purely linear and $\beta = 1.0$ implies the data is purely nonlinear. The confounder ratio is set to 0.3. The number of cycles is randomly set.

**Scaling with Number of Nodes**   We fix the confounder ratio at 0.4. The total number of nodes in the graph is varied from 10 to 80. As in the previous setup, training data consists of combination of observational data and single-node interventions across all nodes, with 500 samples per interventional setting.

**Scaling with Interventions**   We fix $d = 10$ and set the confounder ratio to 0.4. The number of interventions during training varies from 0 to $d$. Zero interventions corresponds to observational data. When fewer than $d$ interventions are provided, the intervened nodes are selected arbitrarily. Each interventional setting consists of 500 samples.

**Non-Contractive SEM**   In this case, we explicitly enforce a non-contractive causal mechanism $\mathbf{F}$ by rescaling edge weights to ensure that the Lipschitz constant of the edge weight matrix exceeds one. We set $d = 10$ and provide observational data and single-node interventions across all nodes, with 500 samples per intervention. The confounder ratio varies between 0.2 and 0.8.

**Scaling with Training Samples**   To examine the sample requirements of DCCD-CONF, we set the confounder ratio to 0.4. Training data consists of observational data and single-node interventions over all nodes, while the number of samples per intervention is varied from 500 to 2500.

**Scaling with outgoing edge density**   In this case, the outgoing edge density of the ER random graphs is varied from 1 to 4. The confounder ratio is set to 0.4 and the number of nodes $d = 10$. The training data consists of observational data and single node experiments over all the nodes in the graph.

**Scaling with noise standard deviation**    In this setting, we vary the maximum noise standard deviation between 0.2 and 0.8. The confounder ratio is set to 0.4 and the number of nodes $d = 10$. The training data consists of observational data and single node experiments over all the nodes in the graph.

**Evaluation Metrics**    Across all experiments, we use Structural Hamming Distance (SHD) to evaluate the accuracy of the estimated directed edges relative to the ground truth. SHD measures the number of modifications (edge additions, reversals, and deletions) required to match the estimated graph to the ground truth. For DCCD-CONF and NODAGS-Flow we fix a threshold value of 0.8 for the estimated adjacency matrix. The recovery of bidirectional edges is assessed using the F1 score, which is defined as:

$$\text{F1 score} = 2\frac{\text{precision} \times \text{recall}}{\text{precision} + \text{recall}}$$

where

$$\text{precision} = \frac{TP}{TP + FP}, \quad \text{recall} = \frac{TP}{TP + FN}$$

and TP, FP, FN denote true positives, false positives, and false negatives, respectively. We use a threshold of 0.01 for the estimated covariance matrix to identify the bidirectional edges.

Additionally, we also measure the performance of DCCD-CONF and the baselines using *Area Under Precision-Recall Curve* (AUPRC) as the error metric. AUPRC computes the area under the precision-recall curve evaluated at various threshold values (the higher the better).

### B.4.2    Gene Perturbation Data set

The dataset was obtained from the Single Cell Portal of the Broad Institute (accession code SCP1064). Following the experimental setup of Sethuraman et al. [27], we filtered out cells with fewer than 500 expressed genes and removed genes expressed in fewer than 500 cells. Due to computational constraints, we selected a subset of 61 perturbed genes (Table 2) from the full genome. The three experimental conditions—co-culture, IFN-$\gamma$, and control—were partitioned into separate datasets, and models were trained and evaluated on each condition independently.

Table 2: The list of chosen genes from Perturb-CITE-seq dataset [55].

| ACSL3 | ACTA2 | B2M | CCND1 | CD274 | CD58 | CD59 | CDK4 | CDK6 |
|---|---|---|---|---|---|---|---|---|
| CDKN1A | CKS1B | CST3 | CTPS1 | DNMT1 | EIF3K | EVA1A | FKBP4 | FOS |
| GSEC | GSN | HASPIN | HLA-A | HLA-B | HLA-C | HLA-E | IFNGR1 | IFNGR2 |
| ILF2 | IRF3 | JAK1 | JAK2 | LAMP2 | LGALS3 | MRPL47 | MYC | P2RX4 |
| PABPC1 | PAICS | PET100 | PTMA | PUF60 | RNASEH2A | RRS1 | SAT1 | SEC11C |
| SINHCAF | SMAD4 | SOX4 | SP100 | SSR2 | STAT1 | STOM | TGFB1 | TIMP2 |
| TM4SF1 | TMED10 | TMEM173 | TOP1MT | TPRKB | TXNDC17 | VDAC2 | | |

## C    Additional Results

### C.1    Additional synthetic experiments

**Experiments on Non-contractive DAGs.**    We evaluated the performance of DCCD-CONF and baseline methods on non-contractive SEMs, where the ground truth DMG is acyclic. While DCCD-CONF assumes a contractive causal mechanism, we adapt it for non-contractive settings using the preconditioning trick proposed by Sethuraman et al. [27]. This approach introduces a learnable diagonal preconditioning matrix $\mathbf{\Lambda}$ , transforming the causal mechanism as follows:

$$\hat{\mathbf{g}}_x = \mathbf{\Lambda}^{-1} \circ \mathbf{g}_x \circ \mathbf{\Lambda},$$

where $\mathbf{g}_x$ , as defined in (8), remains contractive (see Sethuraman et al. [27] for details). We vary the confounder ratio, and the results are summarized in Figure 6a. As shown in Figure, DCCD-CONF effectively learns the ADMG even in non-contractive SEM settings, demonstrating competitive performance against the baselines.

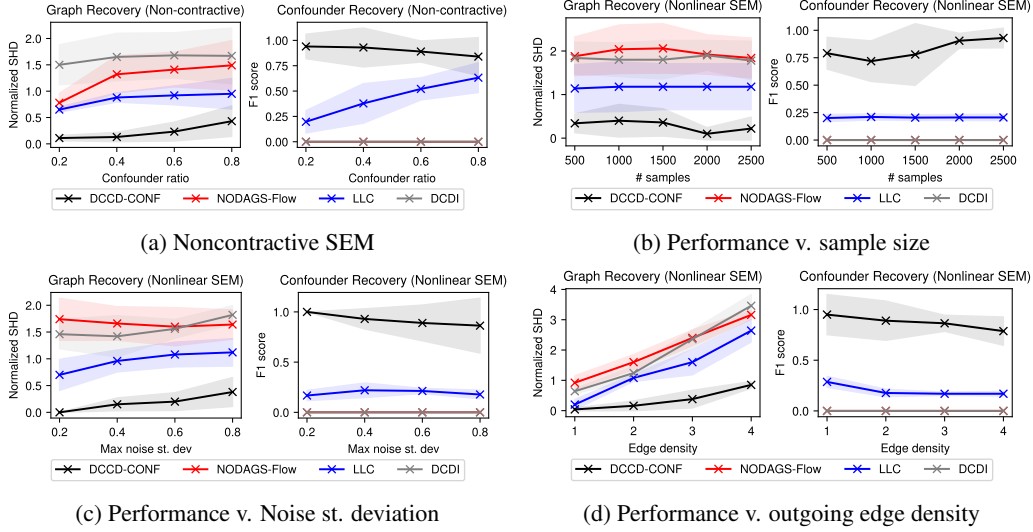

Figure 6: Performance comparison between DCCD-CONF and baseline methods on causal graph recovery and confounder identification, evaluated across varying model parameters. Each subplot details a specific experimental setting.

**Performance comparison vs. sample size**    We also assess the sample requirements of DCCD-CONF. Figure 6b summarizes the results obtained by varying the number of samples per intervention. As shown in the figure, when the confounder ratio is 0.4, DCCD-CONF achieves low SHD even with 500 samples per intervention and attains near-perfect accuracy from 2000 samples onward. However, performance declines slightly as the confounder ratio increases.

**Performance comparison vs. max endogenous noise st. deviation**    In this setting, we compare DCCD-CONF with the baseline by varying the maximum standard deviation of the endogenous noise terms between 0.2 and 0.8, the results are summarized in Figure 6c. DCCD-CONF outperforms the baselines for all noise standard deviations. However, the performance of the models does deteriorate slightly as the noise standard deviation increases.

**Performance comparison vs. outgoing edge density**    In this case, the expected number of outgoing edges from each node is varied between 1 and 4. This affects the sparsity of the resulting graph. The results are summarized in Figure 6d. As seen from Figure 6d, DCCD-CONF still outperforms the baselines, even though the performance of all the models worsens as the edge density increases.

## C.2    Additional performance metrics

In addition to SHD for directed edge recovery and F1 score for bi-directional edge recovery. We also AUPRC to compare DCCD-CONF and the baseline for all of the experimental settings stated in Appendix B.4.1. The results are summarized in Figure 7. Overall, DCCD-CONF performs better than LLC on nonlinear SEMs across all the settings, while achieving perfect AUPRC scores in several cases.

## C.3    Comparison with FCI-JCI

Here, we compare the performance of DCCD-CONF with FCI-JCI [29], which is an extension of FCI algorithm that is capable of handling multiple contexts (in this case interventional settings). FCI-JCI outputs a *Partial Ancestral Graph* (PAG), which is a graph structure that represents the equivalence class of MAGs. We define a modified SHD score in order to check if the DMG estimated by DCCD-CONF belongs to the same equivalence class of the ground truth DMG. To that end, we convert the ground-truth DMG and the estimated DMG to their augmented DMGs and then construct the MAG of the acyclified version of the augmented DMGs. The modified SHD score then computes the discrepancies in the conditions of Theorem A.19, i.e., we count: (i) the

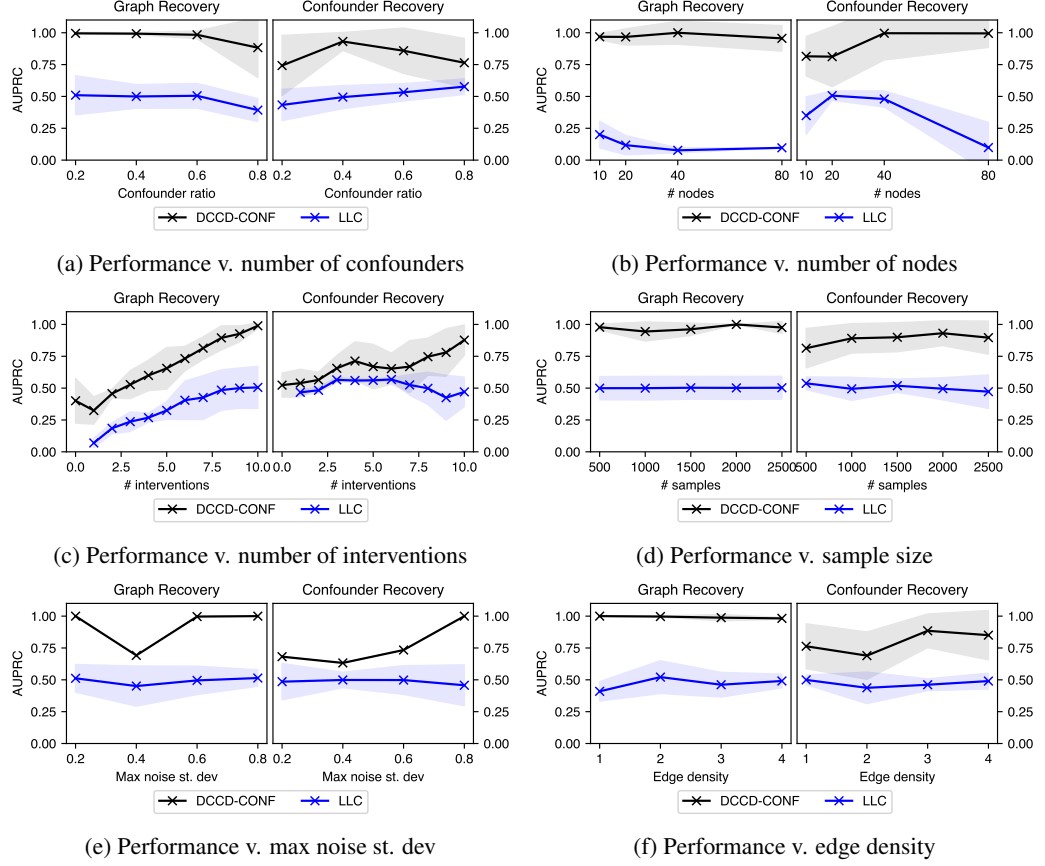

(a) Performance v. number of confounders

(b) Performance v. number of nodes

(c) Performance v. number of interventions

(d) Performance v. sample size

(e) Performance v. max noise st. dev

(f) Performance v. edge density

Figure 7: Comparison of DCCD-CONF and baseline methods on causal graph recovery and confounder identification, measured using AUPRC across varying model parameters.

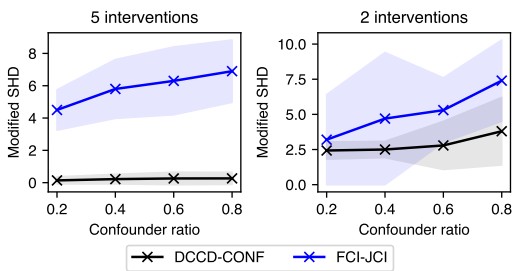

Figure 8: Performance comparison between DCCD-CONF and FCI-JCI with respect to modified SHD as the error metric. The number of observed nodes was set to $d = 5$. For the left plot, all single node interventions along with the observational data were provided as training data. For the right plot, observational data and interventions over two of the nodes (randomly chosen) were used as training data.

number of extra edges ($N_1$) using the skeletons of the estimated MAG and ground truth MAG, (2) number of mismatched unshielded colliders ($N_2$), and (3) discrepancies in the discriminating paths ($N_3$). Similarly, for FCI-JCI, we count the disagreements between the ground-truth MAG and the estimated PAG, i.e., mismatch in skeleton ($N_1$), mismatch in unshielded colliders ($N_2$), invariant edge orientation discrepancies ($N_3$). Finally, the *modified SHD* $= N_1 + N_2 + N_3$. We compare DCCD-CONF and JCI-FCI over two different settings: (i) the training data consists of observational data and single node interventions over all the nodes in the graph, and (ii) the training data consists of observational data and interventions over 2 nodes (randomly chosen) in the graph. Due to the

complexity of computing the modified SHD (as it involves iterating over the discriminating paths and inducing paths) we fix the number of observed nodes to be $d = 5$. In the both the cases, the confounder ratio is varied between 0.2 and 0.8, and nonlinear SEM is used to generate the data. The results are summarized in Figure 8.

As seen from Figure 8, DCCD-CONF outperforms FCI-JCI in the both the settings. However, the performance does decrease as the number of training interventions reduces. We attribute this to the increase sample requirements as the number of training interventions goes down.

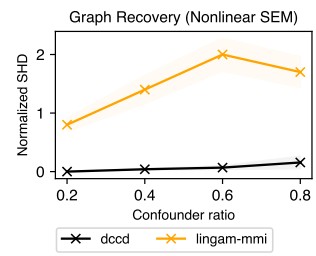

### C.4    Comparison with LiNGAM-MMI

We compare the performance of DCCD-CONF with LiNGAM-MMI [33], which extends the ICA-based LiNGAM [34] to handle hidden confounding. Since LiNGAM-MMI requires iterating over all possible node permutations, its computational cost grows rapidly with the number of nodes; hence, we restrict our comparison to $d = 5$. The training data include both observational samples and

Figure 9:    Performance comparison between DCCD-CONF and LiNGAM-MMI on $d = 5$ node graphs.

single-node interventions for all nodes, with 500 samples per interventional setting. We vary the confounder ratio (i.e., the ratio of confounders to observed nodes) between 0.2 and 0.8. As shown in Figure 9, DCCD-CONF consistently outperforms LiNGAM-MMI across all tested confounder ratios.

### C.5    Hyperparameter Sensitivity

We evaluate the sensitivity of DCCD-CONF to its hyperparameters: (i) the directed edge sparsity regularization coefficient $\lambda_c$, and (ii) the bidirected edge sparsity regularization coefficient $\rho$. Figure 10 summarizes the results for $\lambda_c, \rho \in 0.1, 0.01, 0.001$ on graphs with $d = 10$ nodes and a confounder ratio of 0.3. The model was trained using both observational data and all single-node interventions. As shown in the figure, DCCD-CONF remains fairly robust to hyperparameter variations, achieving normalized SHD values below 0.3 and F1 scores above 0.65 for most combinations of $\lambda_c$ and $\rho$.

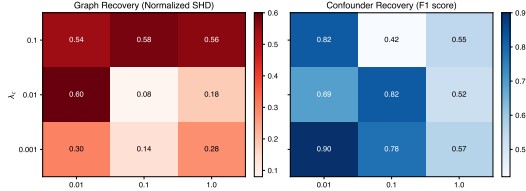

Figure 10: Illustration of DCCD-CONF performance for various choices of hyperparameters $\lambda_c$ (directed edge sparsity regularization coeff.) and $\rho$ (bidirected edge sparsity regularization coeff.).

### C.6    Training Time Comparison

Figure 11 compares the training times of DCCD-CONF and baseline methods. Unlike the other algorithms, LLC does not require stochastic gradient-based training, as it relies solely on solving a series of linear regressions. Consequently, LLC is considerably faster, as shown in Figure 11. Excluding LLC, NODAGS-Flow is the most efficient in terms of runtime; however, it cannot account for confounders within its framework. DCCD-CONF achieves training times comparable to ADMG and DCDI while simultaneously handling confounders, cycles, and nonlinear dependencies. The training times are computed on $d = 10$ node graphs with training dataset consisting of observational and all single-node interventions. DCCD-CONF was trained for 200 epochs. All models were run on RTX6000 GPUs.

### C.7    Additional Results on Perturb-CITE-seq Dataset

In addition to test-set NLL, we evaluate the performance of DCCD-CONF and the baselines on the Perturb-CITE-seq dataset [55] using *Interventional Mean Absolute Error* (I-MAE) as the evaluation

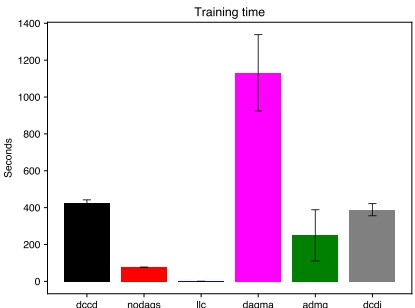

Figure 11: Training time comparison between DCCD-CONF and the baselines

metric. I-MAE is computed as the mean of $\|\boldsymbol{x} + \mathbf{g}_x(\boldsymbol{x})\|_1/d$ over all observations $\boldsymbol{x}$ in the held-out test set. Beyond the baselines discussed in Section 4, we also include Bicycle [56] and DCDFG [47] for comparison. Bicycle supports nonlinear and cyclic structures, whereas DCDFG restricts the search space to acyclic graphs. The results, summarized in Table 3, show that DCCD-CONF remains competitive with state-of-the-art methods under the I-MAE metric.

Table 3: Results on Perturb-CITE-seq [55] gene perturbation dataset. The table presents the average *Mean Absolute Error* (MAE) on the test set, averaged over multiple trials (standard deviation is reported within paranthesis).

| Method | Control | Co-Culture | IFN-$\gamma$ |
|---|---|---|---|
| DCCD-CONF | 0.781 (0.037) | 0.765 (0.046) | 0.843 (0.035) |
| NODAGS | 0.847 (0.018) | 0.762 (0.018) | 0.861 (0.023) |
| Bicycle | 0.782 (0.042) | 0.735 (0.036) | 0.883 (0.028) |
| DCDFG | 0.845 (0.066) | 0.774 (0.038) | 0.891 (0.041) |

Additionally, we report the adjacency matrix of the recovered causal graph for the cell condition "Co-Culture" in Figure 12. DCCD-CONF identified 38 feedback cycles. This number validates prior work showing that gene regulatory networks are rich in feedback loops [55].

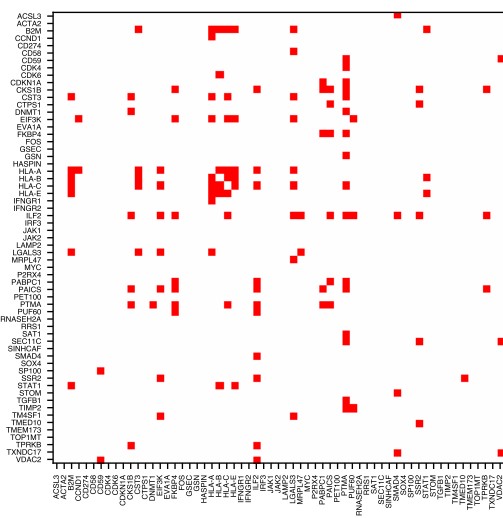

Figure 12: Adjacency matrix of learnt by DCCD-CONF for "Co-Culture" cell condition of Perturb-CITE-seq dataset.

## C.8 Protein Signaling Dataset

We further evaluate DCCD-CONF on a biological dataset for protein signaling network discovery [1], which is widely used as a benchmark for causal discovery algorithms. The dataset contains continuous measurements of multiple phosphory-lated proteins and phospholipid components in human immune system cells, with the corresponding network capturing the ordering of interactions among pathway components. Based on $n = 7466$ samples across $m = 11$ cell types, Sachs et al. [1] identified 20 edges in the underlying graph. Using the consensus network from [1] as ground truth, we evaluate performance using the Structural Hamming Distance (SHD) as the error metric. The results, summarized in Table 4, show that DCCD-CONF performs comparably to the baselines. The recovered directed graph is visualized in Figure 13.

Table 4: Performance comparison on Sachs et al. [1] protein signaling dataset.

| Method | SHD |
|---|---|
| DCCD-CONF | **18** |
| DAG-GNN [10] | 19 |
| DAGMA | 21 |
| NOTEARS | 22 |

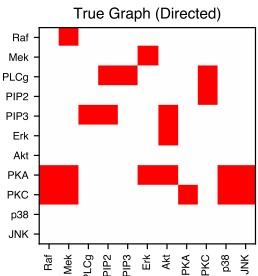 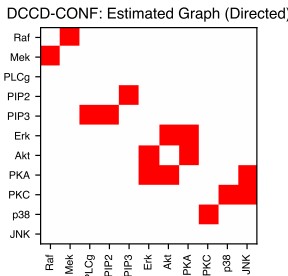

Figure 13: (Left) Consensus graph from Sachs et al. [1], (right) graph learned by DCCD-CONF.

