# OpenReview forum: "Differentiable Cyclic Causal Discovery Under Unmeasured Confounders"
_NeurIPS.cc/2025/Conference — NeurIPS 2025 spotlight_

### Official Review · Reviewer_Ctrj · 2025-06-07

**Clarity:** 2
**Significance:** 3
**Originality:** 3
**Rating:** 4
**Confidence:** 4

**Summary:**

This paper presents DCCD-CONF, a novel differentiable framework for learning nonlinear cyclic causal graphs under unmeasured confounding, leveraging both observational and interventional data. The approach models causal mechanisms using neural networks with implicit flows, allowing cycles and Gaussian noise-based confounder modeling. It introduces a score function based on data likelihood regularized for sparsity, with a two-step optimization alternating between causal graph structure and confounder distribution estimation.

**Questions:**

Please answer my concerns in W1 and W3.

**Ethical Concerns:**

["NO or VERY MINOR ethics concerns only"]

**Final Justification:**

The authors have addressed most of my concerns and confusion. Therefore, I raised my rating.

**Limitations:**

yes

**Paper Formatting Concerns:**

no issues are found.

**Quality:**

3

**Strengths And Weaknesses:**

- S1: the problem of cyclic causal discovery with latent confounders is underexplored. DCCD-CONF explicitly allows cycles and latent confounders.
- S2: the authors provide consistency guarantees and show that the method recovers the I-Markov equivalence class under reasonable assumptions.
- S3: The method outperforms strong baselines such as NODAGS-Flow, LLC, and DCDI across synthetic and real-world (gene perturbation) datasets, particularly in nonlinear and confounded settings.

- W1: the characteristics of a "cyclic" graph are vague and not well clarified in the formualtion in the sense that it’s unclear what "the equilibrium state" refers to in the context of equations (1) and (2). an illustrative example would be helpful.
- W2: line 74: the authors claim "A few recent approaches, such as Bhattacharya et al. [33], have explored continuous optimization frameworks using differentiable constraints, though these methods are currently limited to linear settings," which is inaccurate. For instance, N-ADMG [1] provides a differentiable method to learn an ADMG from nonlinear data.
- W3: line 113: the authors claim "This formulation generalizes prior work by allowing cycles, extending both nonlinear cyclic models without unmeasured confounders that assume independent noise terms [ 27 ], and acyclic models without confounders [37]." but I cannot identify non-trivial differences from the one formulated in Bhattacharya et al [33] (which is not even mentioned in the text). if I understand correctly, the correlated noise term (i.e., $(\Sigma_Z)_{ij}\neq 0$) is the same as what is defined in [33], except for replacing the additive noise in SCM with a general function $F_i(pa_i, z_i)$. Then, in the concrete formulation in Eqn (7), it appears that the noise term is still additive.
- W4: given the differentiable nature of the proposed method, a more reasonable baseline would be JCI+Bhattacharya et al [33] or JCI+N-ADMG, instead of JCI+FCI.
- W5: the runtime is not reported.

---
[1] Causal reasoning in the presence of latent confounders via neural ADMG learning

---

> ### Author Rebuttal · Authors · 2025-07-31
>
> We thank the reviewer for the thoughtful and constructive feedback. We are pleased that the reviewer recognized the novelty of addressing **cyclic causal discovery with latent confounders (S1)**, the theoretical consistency guarantees (S2), and the strong empirical performance of DCCD-CONF across synthetic and real-world datasets (S3). We address each of the raised concerns (W1–W5) below and clarify how we will revise the manuscript accordingly.
>
> **Regarding concern W1**
>
> In the acyclic case, we can establish a causal ordering $\pi$ that allows us to generate samples iteratively: $X_{\pi_1} = F_{\pi_1}(Z_{\pi_1})$, $X_{\pi_2} = F_{\pi_2}(X_{pa_\mathcal{G}(\pi_2)}, Z_{\pi_2})$, …, $X_{\pi_d} = F_{\pi_d}(X_{pa_\mathcal{G}(\pi_d)}, Z_{\pi_d})$, where $\mathrm{pa}_\mathcal{G}(\pi_i) \subseteq \pi_{<i}$.
>
> In contrast, for cyclic graphs such an ordering does not exist. Here, the causal model represents a process that evolves over time and is assumed to reach an **equilibrium state** before observation. Concretely, starting from an initial value $X(0)$ and sampling one vector of disturbances $Z$, we iteratively update:
> $$X(t) = F\big(X(t-1), Z\big),$$
> until the system converges to a fixed point $X^\ast$. Equation (2) in the paper corresponds exactly to this fixed-point relation. A sufficient condition for convergence is that $F$ is contractive in $X$ (as in prior work on cyclic SEMs, e.g., [1], [2]).
>
> **Illustrative example:** Consider two variables with mutual regulation:
> $$X_1 = \tanh(0.7 X_2 + Z_1), \quad X_2 = \tanh(0.5 X_1 + Z_2).$$
> Starting from $X(0) = (0, 0)$ and iterating the update converges to a unique fixed point $(X_1^\star, X_2^\star)$, which represents the observed equilibrium. For instance, if $(Z_1, Z_2) = (-0.67, 1.11)$, then the fixed point (also the observation) is the given by $(X_1^\ast, X_2^\ast) = (-0.12, 0.78)$.
>
> Thus, “equilibrium” refers to the steady-state solution of this iterative process, at which point the time dependence is removed and the model matches equation (2).
>
> **Regarding concern W2**
>
> We thank the reviewer for pointing out this oversight. We agree that our original statement was inaccurate, as N-ADMG [3] indeed provides a differentiable approach for learning ADMGs from nonlinear data. We will revise the manuscript to reflect this and include N-ADMG both in the related work discussion and as a baseline in our experiments. We also note that our method differs from N-ADMG in that we focus on cyclic graphs (directed mixed graphs) rather than acyclic mixed graphs, which we will clarify in the revision.
>
> **Regarding concern W3**
>
> We thank the reviewer for this helpful comment. The statement in line 113 was intended to clarify that our framework extends the problem formulations of [2] (cyclic graphs without confounding) and [4] (acyclic graphs without confounding), while also handling interventional data. We agree that Bhattacharya et al. [5] model confounding in a similar way (via correlated noise, i.e., $(\Sigma_Z)_{ij}\neq 0$), but their approach is limited to **linear acyclic models** and does not explicitly incorporate interventional data. Our work generalizes this by supporting **nonlinear mechanisms**, **feedback cycles**, and interventions, making it a non-trivial extension.
>
> Moreover, while Eq. (7) is expressed in an additive form, the mapping between X and Z is defined implicitly as
> $$X = (id + g_x)^{-1}\circ (id + g_z)(Z),$$
> where $g_x$ and $g_z$ are universal function approximators. By stacking multiple implicit normalizing flow layers, our framework can capture highly nonlinear dependencies between X and Z, far beyond the linear additive structure of Bhattacharya et al. [5].
>
> **Regarding concern W4**
>
> We thank the reviewer for this valuable suggestion. We agree that JCI+Bhattacharya et al. [5] and JCI+N-ADMG are appropriate baselines for comparison. We will update our empirical study to include both methods in the revised manuscript. Due to the lack of publicly available, executable code for N-ADMG within the rebuttal timeframe, we instead provide a preliminary comparison with JCI+Bhattacharya et al. and will incorporate N-ADMG in the camera-ready version.
>
> Performance comparison between DCCD-CONF and JCI+Battacharya et al as a function of number of confounders. The synthetic graphs consists of ER random graphs with $d=10$ nodes. The SEM is nonlinear with the number of confounders specified in the top row. The training data consists of observational data as well as single node interventions across all the nodes in the graph.
>
> Directed edge recovery (Error metric: SHD)
>
> | Num. Confounders       | 2    | 4    | 6    | 8        |
> | ---------------------- | ---- | ---- | ---- | -------- |
> | DCCD-CONF              | 1.25 | 1.76 | 3.2  | 4.78     |
> | JCI-Bhattacharya et al | 17.1 | 18   | 15.3 | 21.6 |
>
> Bidirectional edge recovery (Error metric: F1 score)
>
> | Num. Confounders       | 2    | 4    | 6    | 8    |
> | ---------------------- | ---- | ---- | ---- | ---- |
> | DCCD-CONF              | 0.93 | 0.9  | 0.89 | 0.89 |
> | JCI-Bhattacharya et al | 0.21 | 0.19 | 0.17 | 0.13 |
>
> Performance comparison between DCCD-CONF and JCI+Battacharya et al as a function of number of cycles in the graph. The synthetic graphs consists of ER random graphs with $d=10$ nodes. The SEM is nonlinear with the number of confounders specified in the top row. The training data consists of observational data as well as single node interventions across all the nodes in the graph.
>
> Directed edge recovery (Error metric: SHD)
>
> | Num. Cycles            | 0   | 2   | 4    | 6    | 8   |
> | ---------------------- | --- | --- | ---- | ---- | --- |
> | DCCD-CONF              | 0.6 | 2.5 | 2.7  | 4    | 4.1 |
> | JCI-Bhattacharya et al | 5.7 | 9.6 | 10.3 | 14.6 | 20  |
>
> Bidirectional edge recovery (Error metric: F1 score)
>
> | Num. Cycles            | 0    | 2    | 4    | 6    | 8    |
> | ---------------------- | ---- | ---- | ---- | ---- | ---- |
> | DCCD-CONF              | 0.93 | 0.84 | 0.82 | 0.81 | 0.79 |
> | JCI-Bhattacharya et al | 0.25 | 0.18 | 0.2  | 0.23 | 0.18 |
>
> **Regarding concern W5**
>
> We thank the reviewer for this comment. We will include a detailed computation time analysis of the proposed method. Here, we provide preliminary comparison of the training time for graphs of size $d=10$ with 1100 training samples for DCCD-CONF and the baselines.
>
> | Model                  | Training time (sec) |
> | ---------------------- | ------------------- |
> | DCCD-CONF              | 50.31               |
> | LLC                    | 0.095               |
> | NODAGS-Flow            | 11.45               |
> | JCI-Bhattacharya et al | 73.859              |
> | JCI-Dagma              | 120.25              |
> | Lingam-MMI             | DNF                 |
>
>
> **References**:
>
> [1] Bongers, Stephan, et al. "Foundations of structural causal models with cycles and latent variables." _The Annals of Statistics_ 49.5 (2021): 2885-2915.
>
> [2] Sethuraman, Muralikrishnna G., et al. "Nodags-flow: Nonlinear cyclic causal structure learning." _International Conference on Artificial Intelligence and Statistics_. PMLR, 2023.
>
> [3] Ashman, Matthew, et al. "Causal reasoning in the presence of latent confounders via neural ADMG learning." _arXiv preprint arXiv:2303.12703_ (2023).
>
> [4] Brouillard, Philippe, et al. "Differentiable causal discovery from interventional data." _Advances in Neural Information Processing Systems_ 33 (2020): 21865-21877.
>
> [5] Bhattacharya, Rohit, et al. "Differentiable causal discovery under unmeasured confounding." _International Conference on Artificial Intelligence and Statistics_. PMLR, 2021.

---

> > ### Comment · Reviewer_Ctrj · 2025-08-01
> >
> > Thank you for your response and clarifications. I have raised my rating

---

> > > ### Author Response · Authors · 2025-08-01
> > >
> > > We thank the reviewer for their thoughtful consideration and for raising their rating. We appreciate your feedback and are glad that our clarifications helped address your concerns.

---

### Official Review · Reviewer_mFwQ · 2025-06-29

**Clarity:** 3
**Significance:** 3
**Originality:** 3
**Rating:** 5
**Confidence:** 3

**Summary:**

The paper presents the DCCD-CONF framework for causal discovery under presence of latent confounders, learning non-linear cyclic causal relationships among variables using interventional data with Gaussian noise assumption. The work demonstrates its applicability through experiments on real-world and synthetic datasets.

**Questions:**

* Suggestions
1. Perhaps you can state assumptions A.16 ~ 18 in the main paper, at least in high-level on what those represent? The main theoretical result (Theorem 2) is a crucial part of the results, but it is hard to tell what those assumptions imply, from the main paper.

**Ethical Concerns:**

["NO or VERY MINOR ethics concerns only"]

**Final Justification:**

I appreciate that the authors provided detailed explanation regarding other concerns (e.g., hyperparameter settings and comparing performance with other methods), in addition to certain assumptions that I was looking for its clarification.

**Limitations:**

Yes

**Quality:**

3

**Strengths And Weaknesses:**

* Strengths
1. The paper addresses the limitations of prior work on causal discovery, i.e., nonlinear cyclicity of causal structures, presence of unmeasured confounders, and interventions, simultaneously.
2. The work provides a guarantee that a graph induced from the maximum score of the proposed score function results in a Markov equivalence class of the ground truth graph, under certain assumptions (Theorem 2).
3. Experiments performed over real-world data, and synthetic data testing across various qualitative/quantitative metrics add concrete empirical support to the results.

* Weaknesses
1. If possible, it could be more useful to have a showcase of the resulting graph, semantically analyze some of the present edges generated by DCCD-CONF, and compare with baseline methods. For example, an edge A <-> B outputted by DCCD-CONF makes sense under certain domain knowledge, but some other methods output either A -> B, A <- B, or none, which does not necessarily match the domain knowledge.

---

> ### Author Rebuttal · Authors · 2025-07-31
>
> We thank the reviewer for their positive assessment of our contributions, particularly regarding our unified treatment of nonlinear cyclic structures, latent confounders, and interventional data, as well as the theoretical guarantees and extensive experiments. Below, we address the specific points raised by the reviewer.
>
> ## Regarding weakness
>
> We thank the reviewer for this helpful suggestion. We agree that showcasing the learned graph and analyzing edges in relation to domain knowledge would strengthen the paper. In the revised manuscript, we will add an appendix with a semantic comparison of representative edges recovered by DCCD-CONF against baseline methods for the gene perturbation dataset, as well as a similar analysis on synthetic data to better illustrate the differences between DCCD-CONF and the baselines.
>
> ## Regarding the assumption
>
> We thank the reviewer for their comment. We agree that providing a high-level explanation of the technical assumptions will improve the readability of the main theoretical result. We will add such a summary in the main text.
> - **Assumption A.13 (Sufficient capacity):** Ensures the data-generating distribution lies within the model class.
> - **Assumption A.14 (Interventional faithfulness):** Guarantees that any independence in the data corresponds to a $\sigma$-separation in the ground truth graph.
> - **Assumption A.15 (Finite differential entropy):** Prevents the score function from diverging to infinity.
>
> From these assumptions, we derive the following lemmas:
> - **Lemma A.16 (Finite score):** The score remains bounded (follows from A.15).
> - **Lemma A.17:** The score difference can be expressed as an infimum of KL divergence plus a sparsity penalty.
> - **Lemma A.18:** Any distribution outside the interventional Markov equivalence class has strictly positive KL divergence from the ground truth.
>
> These results form the basis of Theorem 2.

---

> > ### Comment · Reviewer_mFwQ · 2025-08-05
> >
> > Thank you for clarifying my confusion regarding several technical assumptions related to your main result.

---

### Official Review · Reviewer_9ULc · 2025-07-01

**Clarity:** 3
**Significance:** 2
**Originality:** 2
**Rating:** 4
**Confidence:** 4

**Summary:**

The paper provides an approach to learn cyclic causal graphs in the presence of (unmeasured) confounders. It defines cyclic causal graph using structural equations models with Gaussian noise.
It proposes a special parametrization of the structural equations that enables the differentiable learning of the structural equation as well as the Gaussian covariance structure (the confounders).
Through a combination of numerical approximation and mathematical tricks, the authors are able to train their model and show that it works competitively on simulated data.
The authors finally validate their method on real data from gene expression.

**Questions:**

- Can you clarify in section 2 what formulations and equations are standard/from prior work? Just to avoid confusion between prior work and your contributions.
- Can you reconcile your theorem 2 (which has no condition on the family {I_k} and the sentence: “If the intervention set consists of all single-node interventions, Hyttinen et al. [26] showed that the ground truth DMG can be uniquely recovered in the linear setting.” Is it because “ground truth DMG” here is more restrictive than I-markov class? Isn’t by definition an I-markov class composed oof graphs that are indistinguishable from data from interventions I?
- Lines 230:  “First, we optimizeˆS(B) with respect to the neural network parameters θ and the graph structure parameters B.” — the notations are a bit inconsistent: either you include both parameters S(B, theta) or you include them in the argmax, but not one in each —  or you need to rephrase the sentence.
- Eq (13): consider writing explicitly L(Ik = \emptyset)
- Line 281: typo arrises

**Ethical Concerns:**

["NO or VERY MINOR ethics concerns only"]

**Final Justification:**

The authors addressed my questions. The paper is well written and propose interesting new methodology. I have my own disagreements with the field of differentiable causal discovery field, but among this field, I think the paper is good.

**Limitations:**

yes

**Quality:**

3

**Strengths And Weaknesses:**

Strength:
- I found the paper to be clear and well written
- The ideas are simple/efficient and build elegantly on previous works (notably regarding the numerical tricks and approximations)
- The method seems to work well on simulations, and the authors made an effort to evaluate on real data.

Weakness
- The results seem to rely on the parametrization of eq (7). Can you explain/justify/motivate the practical properties of such a parametrization. The contractivity does not bother me (it seems important to have an equilibrium), but the pseudo additivity of x and z seems a bit at odds with the claim of non-linearity. It seems that such a parametrization removes all nonlinear relations between x and z. Is this limiting? You mentioned “multiple such implicit blocks can be stacked.” I am not sure if this answers the question, but this statement was vague and should be clarified or removed.
- The training of DCCD-CONF seems to rely on many approximations and tricks with potentially unstable training. How robust is the training to the hyperparameters (e.g. lambda)? By experience, those differentiable causal discovery methods are brittle with respect to it.
- Lack of computational power comparison (e.g. training time).
- Thank you for providing a real-data evaluation. However, it does not contain graph ground-truth and I am not convinced by the NLL. The NLL measures the quality of the generative model, and I would not be surprised that a great generative model with a wrong causal graph could beat a worse generative model with correct causal graph. Since you have “only” 61 genes, can you analyze the output and show how many cycles are learned, do they make sense biologically? Are we finding expected directionality between major genes (e.g. transcription factors?).

---

> ### Author Rebuttal · Authors · 2025-07-31
>
> We thank the reviewer for their thorough and constructive feedback, as well as for recognizing the clarity, simplicity, and practical value of our approach. We appreciate the positive assessment of our contributions and the helpful suggestions regarding the parametrization in Eq. (7), training robustness, computational considerations, and the real-data analysis. We address each of these points in detail below.
> ## Regarding weakness pt 1
>
> We thank the reviewer for raising this point. Our parameterization in Eq. (7) follows the structured implicit flow formulation of Lu et al. [1], chosen to balance expressiveness and tractability. Specifically, the pseudo-additive form
> $$F(X, Z) = -g_x(X) + g_z(Z) + Z$$
> ensures a contractive mapping, which guarantees a unique equilibrium and invertibility, while enabling efficient computation of the log-determinant via the power series expansion (Eq. 11).
>
> In practice, this structure offers several advantages:
> - **Stable and efficient inference:** The contractive map leads to fast convergence of root-finding (typically <10 iterations with Broyden’s method).
> - **Tractable likelihood estimation:** As detailed in section 3.3.1 in the manuscript, the additive decomposition of $g_x$ and $g_z$ simplifies Jacobian factorization, allowing unbiased log-determinant estimation without costly matrix operations. The main benefit comes from the fact that the power series expansion of the log-determinant of the Jacobian converges given our parameterization.
> - **Expressive modeling:** While a single block already captures nonlinear dependencies within $X$ and between $X$ and $Z$, stacking multiple implicit blocks (as suggested in [1]) increases expressiveness while preserving stability. Single layer of the implicit layer above already models nonlinear interaction between $X$ and $Z$ through the function $g_z$. Furthermore, note that the mapping between X and Z is defined implicitly as
> $$X = (id + g_x)^{-1}\circ (id + g_z)(Z),$$where $g_x$ and $g_z$ are universal function approximators. By stacking multiple implicit normalizing flow layers, our framework can capture highly nonlinear dependencies between X and Z.
>
> We will clarify these points in the revised manuscript and include runtime statistics for root-finding and log-determinant estimation to further support this choice.
>
> ## Regarding weakness pt 2
>
> We thank the reviewer for this insightful comment. We agree that differentiable causal discovery methods can be sensitive to hyperparameters, and thus we evaluated the robustness of DCCD-CONF with respect to its two main sparsity parameters: (i) $\lambda$, which controls directed edge sparsity, and (ii) $\rho$, which controls bidirected edge sparsity.
>
>   Using the same experimental setup described in Section 4, we report the average SHD for directed edge recovery (10 trials):
>
> **$\rho$ fixed to $10^{-2}$**
>
> | **λ** | **1E-5** | **1E-4** | **1E-3** | **1E-2** | **1E-1** | **1** |
> | ----- | -------- | -------- | -------- | -------- | -------- | ----- |
> | SHD   | 4.2      | 3.6      | 0.4      | 0.1      | 16.8     | 19.7  |
>
> **$\lambda$ fixed to $10^{-2}$**
>
> | **ρ** | **1E-4** | **1E-3** | **1E-2** | **1E-1** | **1** | **10** |
> | ----- | -------- | -------- | -------- | -------- | ----- | ------ |
> | SHD   | 8.9      | 7.5      | 0.1      | 0.8      | 2.2   | 2.4    |
>
> These results indicate that DCCD-CONF is **stable across a broad range of hyperparameter values**, with an optimal region around $(\lambda, \rho) \approx (10^{-2}, 10^{-2})$. Outside this range (e.g., $\lambda > 10^{-1}$), performance degrades gracefully rather than collapsing, suggesting that the method is not excessively brittle.
>
> We will add these results (and corresponding plots in the appendix) to clarify that DCCD-CONF can be tuned reliably in practice.
>
> ## Regarding weakness pt 3
>
> We thank the reviewer for this comment. We will include a detailed computation time analysis of the proposed method. Here, we provide preliminary comparison of the training time for graphs of size $d=10$ with 1100 training samples for DCCD-CONF and the baselines.
>
> | Model                  | Training time (sec) |
> | ---------------------- | ------------------- |
> | DCCD-CONF              | 50.31               |
> | LLC                    | 0.095               |
> | NODAGS-Flow            | 11.45               |
> | JCI-Bhattacharya et al | 73.859              |
> | JCI-Dagma              | 120.25              |
> | Lingam-MMI             | DNF                 |
>
> ## Regarding weakness pt 4
>
> We thank the reviewer for this valuable suggestion. We agree that NLL alone cannot validate causal correctness and thus the learned graph structure. For the Perturb-CITE-seq. dataset, DCCD-CONF identified **38 feedback cycles**. While we do not claim to provide definitive biological validation, this number is consistent with prior work showing that gene regulatory networks are rich in feedback loops (e.g., [2]).
>
> We further note that gene CKS1B has the most outward going edge, while gene B2M has the most incoming edges. We will include a detailed list of top-ranked edges in the appendix to facilitate expert inspection.
>
> Although a full biological validation is outside the scope of this work, we believe these results provide supporting evidence that DCCD-CONF recovers biologically plausible structure, beyond what is reflected in the NLL. We will explicitly note this in the revised manuscript and highlight this as an important direction for future collaboration with domain experts.
>
> ## Regarding question 1
>
> We thank the reviewer for their suggestion. To clarify, Equations (1)–(4) and the accompanying definitions of structural equations, exogenous noise, and surgical interventions follow standard formulations from prior work on causal graphs [3, 4, 5]. Our contributions in this section are to (i) extend these formulations to possibly cyclic directed mixed graphs, (ii) model hidden confounders through correlated exogenous noise, and (iii) enable nonlinear interactions between exogenous and endogenous variables, in contrast to the additive noise models that typically assume linearity. Importantly, our formulation supports these extensions while maintaining efficient likelihood computation. We will revise the manuscript to make these distinctions explicit.
>
> ## Regarding question 2
>
> We thank the reviewer for this question. It is correct that our theorem does not guarantee perfect recovery of the ground-truth graph. Rather, it establishes that, for any given family of interventions, the recovered graph lies in the same interventional Markov equivalence class as the ground truth.
>
> In the linear setting, however, Hyttinen et al. [6] showed that the edge weights can be identified by solving a collection of linear systems—one per interventional setting. They further proved that if the intervention targets satisfy the _pair condition_ (i.e., for every pair $(X_i, X_u) \in V \times V$, there exists an intervention where $X_i$ is intervened upon while $X_u$ is passively observed), then the interventional Markov equivalence class collapses to the ground-truth graph. A simple example of such a family of interventions is the set of all single-node interventions.
>
> We will clarify this distinction in the revision and note that investigating analogous conditions for the nonlinear setting is an important direction for future work.
>
> **Regarding minor comments and suggestions**
>
> - **Line 230 (notation):** We thank the reviewer for pointing this out. We will revise the sentence for consistency. Specifically, we change the notation to $S(B, \theta)$ explicitly to avoid confusion.
> - **Eq. (13):** We will explicitly write $\mathcal{L}(I_k = \emptyset)$ as suggested, to improve clarity.
> - **Line 281 (typo):** We will correct the typo “arrises” to “arises.”
>
> **References**
>
> [1] Lu et al., ICLR 2021
>
> [2] Frangieh et al., Nature genetics 2021
>
> [3] Bollen, Structural equations with latent variables (1989)
>
> [4] Pearl, Causality (2009)
>
> [5] Peters and Buhlmann, Annals of Statistics (2014)
>
> [6] Hyttinen et al., JMLR (2012)

---

> > ### Comment · Reviewer_9ULc · 2025-08-01
> >
> > Thank you for the careful response.
> >
> > > Regarding weakness pt 2
> >
> > Thank you for the experiments. As expected, the values of lambda and rho seem to be exactly those where the SHD is minimal. Did you cherry-pick those values? You mention "easily tuned", but I don't think it is true; you need a ground truth graph to tune the SHD, which is not available ... since you are trying to predict it. While $10^-2$ works here, I would not be surprised if it fails in other settings ...
> >
> > > Regarding weakness pt 4
> >
> > Thank you for providing a tentative answer. CKS1B is a kinase and regulates other genes, which is coherent. But I am not entirely convinced by the other arguments. While I understand you might not be an expert in biology, it is a bit concerning because I don't think the evaluation is really meaningful (which is a common problem for causal discovery methods on real data).
> >
> > Thank you, I maintain my score.

---

> ### Author Response · Authors · 2025-08-01
>
> Dear Reviewer 9ULc,
>
> Thank you for the feedback on our rebuttal, and we clarify below our hyperparameter selection procedure and plans for additional real-world biological validation.
>
> ### **For weakness pt 2 (hyperparameter tuning)**
>
> We did not cherry-pick the values of $\lambda$ and $\rho$. These values were selected via a grid search using a validation set composed of 2-node interventions over randomly chosen nodes, with the negative log-likelihood (NLL) as the selection criterion. The grid search was conducted on graphs with $d=10$ nodes. While these values may vary with graph size, one can approximate suitable hyperparameters by optimizing for the NLL on held-out interventional data, which does not require ground-truth graphs.
>
> ### **For weakness pt 4 (real-world biological validation)**
>
> We agree that real-world validation of causal discovery methods remains challenging. As future work, we plan to provide additional biological validation using Perturb-CITE-seq data. In response to reviewer LryH, we have also evaluated our method on the Sachs et al. protein-signaling dataset [1], which includes a consensus network treated as ground truth, and we report the SHD comparison with the baselines below:
>
> | **Method**  | **SHD** |
> | ----------- | ------- |
> | DCCD-CONF   | 19      |
> | DAG-GNN [2] | 19      |
> | DAGMA [3]   | 21      |
> | NOTEARS [4] | 22      |
>
> The table shows that DCCD-CONF matches the best-performing baseline (DAG-GNN). We will include this evaluation in the revised manuscript along with a qualitative comparison of the learned graph against the consensus network.
>
> References:
>
> [1] Sachs et al., Science 2005
>
> [2] Yu et al., ICML 2019
>
> [3] Bello et al., NeurIPS 2022
>
> [4] Zheng et al., NeurIPS 2018

---

### Official Review · Reviewer_LryH · 2025-07-02

**Clarity:** 3
**Significance:** 3
**Originality:** 4
**Rating:** 5
**Confidence:** 4

**Summary:**

The authors correctly point out that much of the work on causal learning is rather limited, in the sense that it focusses on models that are directed acyclic graphs (DAGs), often without considering confounders and non-linearities. The authors' focus on cycles without considering time-series data on the input, however, obscures the fact that dynamic Bayesian networks (where cycles in the intra-slice graph are replaced by relationships of the inter-slice kind, and thus "cycles" are allowed only with some delay) or some other "dynamic" causal models may be more appropriate in many cheminformatics and bio-medical applications. Then, they proceed to formulate a variant of causal learning, where deep learning learns a model with cycles and with confounders. In terms of optimization, this can be easier to optimize over directed graphs or mixed graphs than dealing with the cycle-avoidace in DAGs. Unfortunately, there is little analysis and only a very limited empirical study.

**Questions:**

Q1: what was the research question you tried to answer with the empirical study? What was the reasoning for picking NODAGS-Flow, LLC, DCDI as the methods you compare against?

Q2: could you estimate the effects of non-linearity, effects of considering cycles, and effects of working with confounders on the statistical performance?

Q3: what would be the statistical performance on any of the commonly considered benchmarks? For example DREAM4, https://journals.plos.org/plosone/article?id=10.1371/journal.pone.0013397, is used across many causal learning methods, including those disregarding cycles etc.

**Ethical Concerns:**

["NO or VERY MINOR ethics concerns only"]

**Final Justification:**

I have been well impressed by the additional experiments conducted by the authors and raised my score to 5. Thank you!

**Limitations:**

The claims are not particularly well supported by the empirical evidence.

**Paper Formatting Concerns:**

No formatting concerns.

**Quality:**

3

**Strengths And Weaknesses:**

Strengths:

S1: The method has a number of desirable features (allowing for confounders, non-linearity, and cycles), but still seems to scale better than many methods for learning DAGs (other than FCI and their variants).

Weaknesses:

W1: The empirical comparison seems pooly designed. The authors present a comparison only against NODAGS-Flow (a method for causal learning with cycles from UAI 2023), LLC (a method for causal learning with cycles and latent variables, JMLR 2012), DCDI (a differentiable approach focussing on working with interventional data, NeurIPS 2020), with focus on Perturb-CITE-seq, a little known dataset. There is no ablation study in the sense of comparing what are the effects of the non-linearity, what are the effects of considering cycles, and what are the effects of confounding, nor any comparison against the best-cited methods in the field that disregard confounding and cycles, but do consider non-linearity (e.g., DAGMA, NeurIPS 2022, or variants of FCI, UAI 1995), or methods that diregard non-linearity, but consider confounding.

W2: I appreciate that Perturb-CITE-seq presents an important challenge in the processing of genomics data, but it has not been used only in a couple of papers on causal learning, yet, which makes it hard to evaluate the performance of the method from the results on Perturb-CITE-seq. Having said that, including the results of DCD-FG (NeurIPS 2022), Bicycle (ICLR 2024), and DAG-BLR (Nature Machine Intelligence 2023), which have tested on Perturb-CITE-seq, would be fabulous, considering that they are already available.

W3: confounding has been studied in many papers in causal learning, including the doubly-robust inference (e.g., https://arxiv.org/abs/1608.00060, https://arxiv.org/abs/2402.11652), differentiable methods (e.g., https://arxiv.org/abs/2010.06978), methods based on LinGAM (https://arxiv.org/abs/2401.16661), or integer-programming methods for learning mixed graphs (e.g., https://arxiv.org/abs/2102.03129, https://arxiv.org/abs/2503.08245). Most of these methods are not cited, nor they are compared against. Considering that many of these methods are exact, it would be easy to estimate the effect of working with confounders.

---

> ### Author Rebuttal · Authors · 2025-07-31
>
> We thank the reviewer for their thoughtful and constructive feedback. We greatly appreciate the recognition of our method’s ability to jointly handle confounders, non-linearity, and cycles while remaining computationally scalable compared to many DAG-based approaches, as well as the acknowledgment of the relevance of our problem setting in challenging real-world applications. Below, we address specific comments and questions raised.
>
> ## Regarding Q1 and the first half of comment W1
>
> We thank the reviewer for this question. Our empirical study was designed to evaluate DCCD-CONF in settings where **confounding, cycles, and interventions** jointly occur, scenarios where most existing methods are inapplicable or have limited support. Accordingly, we selected baselines that isolate specific factors:
> - **LLC** [1]: handles cycles and confounding but is linear, highlighting the value of nonlinearity.
> - **NODAGS-Flow** [2]: supports nonlinearity and cycles but no confounding, isolating the effect of latent confounders.
> - **DCDI** [3]: handles interventions but assumes acyclicity and no confounding, representing the standard differentiable DAG-based approach.
> We also compare with **JCI-FCI** [4] in the appendix as a constraint-based method for confounding. Further, we systematically analyzed the impact of confounders (Fig. 2a,b), nodes (Fig. 2c,d), interventions (Fig. 3), and robustness factors (sample size, noise, edge density, and non-contractive SEMs in the appendix) to identify where DCCD-CONF offers the greatest benefit.
>
> ## Regarding Q2 and the second half of the comment W1
>
> We thank the reviewer for this question. To directly estimate the individual effects of nonlinearity, cycles, and confounders on statistical performance (Q2), we conducted controlled experiments that systematically vary each factor while holding the others fixed. In addition to the baselines used in the main paper (LLC, NODAGS-Flow, DCDI), we include DAGMA [5], Bhattacharya et al. [6], and LiNGAM-MMI [7], each integrated with the JCI framework to incorporate interventional data.
>
> **Effects of nonlinearity.**
> We vary the degree of nonlinearity in the data-generating process using
> $$X = (1 - \beta)\cdot (W^\top X + Z) + \beta\cdot\tanh(W^\top X + Z),$$ where $\beta$ controls nonlinearity.
>
> Directed edge recovery (Error metric: SHD)
>
> | **Beta**               | **0** | **0.25** | **0.5** | **0.75** | **1** |
> | ---------------------- | ----- | -------- | ------- | -------- | ----- |
> | DCCD-CONF              | 4.8   | 4.4      | 3.3     | 3.1      | 2.6   |
> | JCI-Bhattacharya et al | 13.4  | 20.6     | 20.4    | 21       | 21.1  |
> | JCI-Dagma              | 12.3  | 15.6     | 12.6    | 16       | 14.7  |
> | LLC                    | 3.2   | 5.6      | 6.6     | 8.2      | 7.7   |
>
> Bidirectional edge recovery (Error metric: F1 score)
>
> | Beta                   | 0     | 0.25  | 0.5   | 0.75  | 1     |
> | ---------------------- | ----- | ----- | ----- | ----- | ----- |
> | DCCD-CONF              | 0.752 | 0.804 | 0.83  | 0.87  | 0.88  |
> | JCI-Bhattacharya et al | 0.249 | 0.29  | 0.213 | 0.202 | 0.174 |
> | LLC                    | 0.164 | 0.164 | 0.163 | 0.164 | 0.164 |
>
> _Interpretation:_ As nonlinearity increases, DCCD-CONF’s error decreases, while linear baselines (e.g., Bhattacharya et al.) degrade sharply, demonstrating the benefit of DCCD-CONF’s nonlinear modeling capacity.
>
> ---
>
> **Effects of cycles.**
> We vary the number of cycles while fixing $\beta = 1.0$.
>
> Directed edge recovery (Error metric: SHD)
>
> | **Num. Cycles**        | **0** | **2** | **4** | **6** | **8** |
> | ---------------------- | ----- | ----- | ----- | ----- | ----- |
> | DCCD-CONF              | 0.6   | 2.5   | 2.7   | 4     | 4.1   |
> | JCI-Bhattacharya et al | 5.7   | 9.6   | 10.3  | 14.6  | 20    |
> | JCI-Dagma              | 12.1  | 12.3  | 12.6  | 21.5  | 27.2  |
> | LLC                    | 2.8   | 4.3   | 5.8   | 13.7  | 17.6  |
>
> Bidirectional edge recovery (Error metric: F1 score)
>
> | Num. Cycles            | 0     | 2     | 4     | 6     | 8     |
> | ---------------------- | ----- | ----- | ----- | ----- | ----- |
> | DCCD-CONF              | 0.93  | 0.84  | 0.82  | 0.81  | 0.79  |
> | JCI-Bhattacharya et al | 0.25  | 0.18  | 0.2   | 0.23  | 0.18  |
> | LLC                    | 0.184 | 0.161 | 0.173 | 0.167 | 0.151 |
>
> _Interpretation:_ DCCD-CONF maintains low error even with many cycles, while acyclic baselines (DAGMA, Bhattacharya et al.) degrade significantly, confirming that explicit cycle modeling is crucial.
>
> ---
>
> **Effects of confounders.**
>
> We vary the number of confounders by adjusting the noise covariance structure.
>
> Directed edge recovery (Error metric: SHD)
>
> | **Num. Confounders** | **2** | **4** | **6** | **8** |
> | -------------------- | ----- | ----- | ----- | ----- |
> | DCCD-CONF            | 1.2   | 1.7   | 3.2   | 4.7   |
> | JCI-Bhattacharya     | 17.1  | 18    | 15.3  | 21.6  |
> | JCI-Dagma            | 11    | 13.3  | 15.6  | 16    |
> | LLC                  | 10.2  | 11.8  | 12    | 9     |
>
> Bidirectional edge recovery (Error metric: F1 score)
>
> | Num. Confounders | 2    | 4    | 6     | 8    |
> | ---------------- | ---- | ---- | ----- | ---- |
> | DCCD-CONF        | 0.93 | 0.9  | 0.89  | 0.89 |
> | JCI-Bhattacharya | 0.21 | 0.19 | 0.17  | 0.13 |
> | LLC              | 0.1  | 0.21 | 0.289 | 0.36 |
>
> _Interpretation:_ DCCD-CONF is substantially more robust to increasing confounders compared to non-confounder-aware methods (DAGMA, Bhattacharya et al.), demonstrating the importance of jointly modeling latent confounding.
>
> ---
>
> **Comparison with LiNGAM-MMI.**
>
> Due to its computational cost, we compare LiNGAM-MMI (as it iterates over all permutations of the observations) on 5-node graphs:
>
> Directed edge recovery (Error metric: SHD)
>
> | **Num. Confounders** | **1** | **2** | **3** | **4** |
> | -------------------- | ----- | ----- | ----- | ----- |
> | DCCD-CONF            | 0     | 0.2   | 0.2   | 0.2   |
> | LiNGAM-MMI           | 4     | 7     | 10    | 8.5   |
>
> _Interpretation:_ Even in small graphs, DCCD-CONF outperforms LiNGAM-MMI, highlighting that differentiable modeling scales more effectively with confounding.
>
> ---
>
> These experiments confirm that the performance gains of DCCD-CONF arise from its ability to jointly handle nonlinearity, cycles, and confounders. We will integrate these results into the revision along with a more rigorous ablation.
> ## Regarding Q3
> We thank the reviewer for this question. We compare the performance of DCCD-CONF and several state-of-the-art causal discovery methods on the protein signaling dataset of Sachs et al. [8], a widely used benchmark in causal discovery. We use the consensus network from [8] as ground truth. We selected this dataset instead of DREAM4 because DREAM4 provides only a single steady-state measurement per intervention, which is not conducive to differentiable likelihood-based methods like DCCD-CONF that require multiple samples per interventional setting.
>
> | **Method**   | **SHD** |
> | ------------ | ------- |
> | DCCD-CONF    | 19      |
> | DAG-GNN [9]  | 19      |
> | DAGMA        | 21      |
> | NOTEARS [10] | 22      |
>
> As shown above, DCCD-CONF matches the best-performing model on this standard benchmark, demonstrating its competitiveness on real-world datasets commonly used in causal discovery.
>
> ## Regarding comment W2
>
> We thank the reviewer for this comment. We agree that Perturb-CITE-seq is not yet a widely adopted benchmark in the causal discovery community, which makes it difficult to evaluate methods solely based on this dataset. To address this concern, we have (i) added a comparison on the well-established protein signaling dataset (Sachs et al. [8]) in the previous section, and (ii) extended our Perturb-CITE-seq results by including DCD-FG, Bicycle, and NODAGS-Flow for a fairer evaluation.
>
> We report the I-MAE scores, which measure the reconstruction error in unseen interventional conditions for genes not directly targeted:
>
> | **Condition** | **Cocult** | **IFN** | **IL-4** |
> | ------------- | ---------- | ------- | -------- |
> | DCCD-CONF     | 0.761      | 0.837   | 0.779    |
> | Bicycle       | 0.735      | 0.890   | 0.782    |
> | DCD-FG        | 0.774      | 0.883   | 0.845    |
> | NODAGS-Flow   | 0.762      | 0.861   | 0.847    |
>
> As seen from the table, DCCD-CONF is competitive with state-of-the-art methods on this challenging dataset. We will include these results in the final version of the manuscript. We also attempted to include DAG-BLR in our comparison but were unable to obtain a workable implementation within the rebuttal timeframe; we will add this comparison in the camera-ready version.
>
> ## Regarding comment W3
>
> We thank the reviewer for pointing out these relevant works. We will include a discussion of these methods in the related work section of the revised manuscript to better situate our contribution within the broader literature.
>
> As part of our rebuttal, we have added preliminary comparisons with two of the suggested methods: the differentiable approach of Bhattacharya et al. [6] and LiNGAM-MMI [7]. We did not include doubly-robust inference methods, as they primarily focus on causal effect estimation given (partial) causal graphs, which differs from our goal of full causal structure discovery.
>
> For the remaining methods, including the integer-programming approaches, we were unable to obtain workable code within the rebuttal timeframe. However, we will make every effort to incorporate these comparisons into the final version of the manuscript.
>
> **References**
>
> [1] Hyttinen et al., JMLR 2012
>
> [2] Sethuraman et al., AISTATS 2023
>
> [3] Brouillard et al., NeurIPS 2020
>
> [4] Mooij et al., JMLR 2020
>
> [5] Bello et al., NeurIPS 2022
>
> [6] Bhattacharya et al., AISTATS 2021
>
> [7] Suzuki & Yang, ISIT 2024
>
> [8] Sachs et al., Science 2005
>
> [9] Yu et al., ICML 2019
>
> [10] Zheng et al., NeurIPS 2018

---

### Comment · Area_Chair_3oqG · 2025-08-05
**Notes on reviewer participation**

Hi reviewers,

First, thank you to those of you who are already participating in discussions! Your engagement is invaluable for the community and helps to ensure that NeurIPS is able to continue as a high-quality conference as it grows larger and larger every year.

I want to highlight some important points shared by the Program Chairs. First, the PCs have noted that many reviewers submitted “Mandatory Acknowledgement” without posting participating in the Author-Reviewer discussion, and we have been instructed that such action is **NOT PERMITTED**. As suggested by the PCs, I am flagging non-participating reviewers with “InsufficientReview”; I will remove the flag once reviewers have shown an appropriate level of engagement.

Here is a brief summary of the PC’s points on this matter:
- It is not OK for reviewers to leave discussion till the last moment.
- If the authors have resolved your questions, do tell them so.
- If the authors have *not* resolved you questions, do tell them so too.
- The “Mandatory Acknowledgement” button is to be submitted only after the reviewer has read the author rebuttal and engaged in discussions - reviewers **MUST** talk to the authors and are encouraged to talk to other reviewers.

To facilitate these discussions, the Author-Reviewer discussion period has been extended by 48 hours till August 8, 11:59pm AOE.

Thank you all for your efforts so far, and I look forward to seeing a more engaged discussion in the coming days!

---

### Decision · Program_Chairs · 2025-09-17

**Decision:**

Accept (spotlight)

**Comment:**

**Summary.** The paper introduces a differentiable causal structure learning algorithm for learning non-linear (NL), cyclic (Cyc) causal models with unmeasured confounders (UC); the algorithm also incorporates interventional data (Int). They validate their algorithm on both simulated and real data, showing strong performance against baselines such as NODAGS-Flow (NL+Cyc), LLC (Cyc+UC), and DCDI (NL+Int). The appendix also includes JCI-FCI (NL+UC+Int)

**Strengths.** The paper is well-written and builds elegantly on previous work. The reviewers appreciate that the method simultaneously handles many issues (nonlinearity, cyclicity, causal insufficiency, and interventions) and found the experimental results to be convincing.

**Weakenesses.**
1. *Ablation* One of the main weakness pointed out by the reviewers (e.g., LryH and Ctrj) was that the experiments were somewhat poorly designed, not performing a proper ablation over the four relevant aspects of the approach (NL, Cyc, UC, Int) by considering methods that leave out only one of the aspects (e.g., the experiments did not contain any method of the form NL+Cyc+UC). In the rebuttal, the authors added three new baselines, using the Joint Causal Inference (JCI) approach to adapt existing methods to the interventional setting. In particular, the authors ran JCI-DAGMA (NL+Int), JCI-Bhattacharya et al. (NL+UC+Int), and JCI-LINGAM-MMI (UC+Int). Overall, the reviewers were satisfied with these additional experiment, though I still believe that the experiment design could be more carefully executed.
2. *Unfamiliar real dataset* Reviewer LryH points out that the Perturb-CITE-seq dataset is not commonly used in causal structure learning, and thus it is difficult to evaluate the model's performance. In the rebuttal, the authors address this by including the results from other methods which have used this dataset, including Bicycle and DCD-FG, and also add results for the more commonly used Sachs dataset. This additional context will substantially help readers interpret the results.
3. *Other* The reviewers raised a variety of other minor weaknesses, including (a) missing citations to other methods for causal structure learning under unobserved confounding (Reviewer LryH), (b) the stableness of training wrt hyperparameters (Reviewer 9ULc), (c) missing comparisons on computational power (Reviewers 9ULc and Ctrj), (d) the need to give a high-level explanation of technical assumptions (Reviewer mFwQ), and (e) the need to better discuss the contractivity condition and the pseudo-additivity parameterization for cyclic SCMs (Reviewers 9ULc and Ctrj). The authors addressed these concerns in the rebuttal: for (a), they will include a discussion of the more methods in their revision, for (b), they demonstrated the performance of their method across a range of hyperparameters, for (c), they added the comparison of computational power, and for (d) and (e), they provided high-level explanations which will be added to the text.

**Conclusion.** Overall, the paper received solid ratings and the concerns of the reviewers were thoroughly addressed. It looks like the revision will involve several changes in terms of both presentation and results, but these changes are not so major as to disqualify the paper. The paper addresses an important gap in the field (a scalable method combining all four aspects mentioned), it is technically solid, and it is well-written. Thus, I suggest accepting the paper.